# High-throughput single nucleus total RNA sequencing of formalin-fixed paraffin-embedded tissues by snRandom-seq

Ziye Xu[1,2,14], Tianyu Zhang[3,14], Hongyu Chen[4,5,6,14], Yuyi Zhu[2], Yuexiao Lv[2], Shunji Zhang[7], Jiaye Chen [8], Haide Chen[2], Lili Yang[9], Weiqin Jiang[10], Shengyu Ni[3], Fangru Lu[3], Zhaolun Wang[3], Hao Yang[3], Ling Dong[3], Feng Chen[9], Hong Zhang[7,11], Yu Chen[1], Jiong Liu[3], Dandan Zhang [12,13], Longjiang Fan [4,5,6 ✉], Guoji Guo [2 ✉] & Yongcheng Wang [1,2,7 ✉]

Formalin-fixed paraffin-embedded (FFPE) tissues constitute a vast and valuable patient material bank for clinical history and follow-up data. It is still challenging to achieve single cell/nucleus RNA (sc/snRNA) profile in FFPE tissues. Here, we develop a droplet-based snRNA sequencing technology (snRandom-seq) for FFPE tissues by capturing full-length total RNAs with random primers. snRandom-seq shows a minor doublet rate (0.3%), a much higher RNA coverage, and detects more non-coding RNAs and nascent RNAs, compared with state-of-art high-throughput scRNA-seq technologies. snRandom-seq detects a median of >3000 genes per nucleus and identifies 25 typical cell types. Moreover, we apply snRandom-seq on a clinical FFPE human liver cancer specimen and reveal an interesting subpopulation of nuclei with high proliferative activity. Our method provides a powerful snRNA-seq platform for clinical FFPE specimens and promises enormous applications in biomedical research.

Routine formalin-fixed paraffin-embedded (FFPE) tissues are the most common archivable specimens, constituting a vast and valuable patient material bank for clinical history, follow-up data, etc[1]. The tissue morphology and cellular details of FFPE tissues are well-preserved for histopathology by the formaldehyde crosslinking among DNA, RNA, and proteins. Inevitably, the irreversible modifications caused by formalin fixation on macromolecules in FFPE samples always make it challenging for molecular biology applications. Recent studies have made great progress in transcription profiling in FFPE samples by optimal RNA extraction methods[2] or spatial in situ profiling[3]. In the last few years, high-throughput single-cell/nuclei RNA sequencing (scRNA/snRNA-seq) methods have revolutionized the entire field of biomedical research[4–6]. We have constructed the first human and mouse cell atlas with our customized high-throughput scRNA-seq platforms[7,8].

[1]Department of Laboratory Medicine, the First Affiliated Hospital, Zhejiang University School of Medicine, Hangzhou, China. [2]Liangzhu Laboratory, Zhejiang University Medical Center, Hangzhou, China. [3]M20 Genomics, Hangzhou, China. [4]School of Medicine, Hangzhou City University, Hangzhou, China. [5]Institute of Bioinformatics, Zhejiang University, Hangzhou, China. [6]James D. Watson Institute of Genome Sciences, Zhejiang University, Hangzhou, China. [7]College of Biomedical Engineering and Instrument Science, Zhejiang University, Hangzhou, China. [8]Department of Biomedical Informatics, Harvard Medical School, Boston, USA. [9]Department of Radiology, the First Affiliated Hospital, Zhejiang University School of Medicine, Hangzhou, China. [10]Department of Colorectal Surgery, the First Affiliated Hospital, Zhejiang University, Hangzhou, China. [11]Department of Nuclear Medicine and PET/CT Center, the Second Affiliated Hospital, Zhejiang University School of Medicine, Hangzhou, China. [12]Department of Pathology, and Department of Medical Oncology of the Second Affiliated Hospital, Zhejiang University School of Medicine, Hangzhou, Zhejiang, China. [13]Department of Pathology, Key Laboratory of Disease Proteomics of Zhejiang Province, School of Medicine, Zhejiang University, Hangzhou, China. [14]These authors contributed equally: Ziye Xu, Tianyu Zhang, Hongyu Chen. ✉e-mail: fanlj@zju.edu.cn; ggj@zju.edu.cn; yongcheng@zju.edu.cn

Accurate transcriptomics characterization of every single cell in clinical FFPE specimens is believed to have the ability to deliver a better understanding of cell heterogeneity and population dynamics, thereby improving the precision diagnostics, treatment, and prognosis of human disease. However, both single intact cell/nuclei isolation and RNA capture from FFPE tissues are still challenging due to RNA crosslinking, modification, and degradation.

Currently, most popular high-throughput sc/snRNA-seq platforms, such as 10X Genomics Chromium Single Cell 3' Solution, rely on oligo(dT) to capture poly(A)+ RNAs, such that mainly matured messenger RNA (mRNA) will be detected rather than non-poly-adenylated RNAs for analysis. In addition, these oligo(dT)-based sc/snRNA-seq methods are primarily restricted to either fresh or fresh-frozen samples, as oligo(dT) primers usually fail on degraded RNAs. Various methods have been developed to overcome these challenges from different perspectives. SMART-seq-total[9] and VASA-seq[10] capture both polyadenylated and non-polyadenylated transcripts by deploying an extra step of tailing all RNA molecules with poly(A). On the other hand, SPLiT-seq[11] was reported to be successfully used in fixed cells using random primer that was more efficient and broader to capture total RNAs[12,13]. However, these methods were not yet workable for FFPE tissues. Two methods that were recently posted on bioRxiv, snPATHO-Seq[14], and snFFPE-seq[15], provided optimized methods to isolate single intact nuclei from FFPE tissues to perform snRNA-Seq, which demonstrates the feasibility of snRNA-Seq in FFPE tissues and unlocks a dimension of these hard-to-use samples. snPATHO-Seq depends on the probe-based 10X Genomics technology so that only limited genes can be detected[14]. snFFPE-seq utilizes the poly(A)-based 10X Genomics platform, which is not sensitive enough to capture the low-quality RNAs from FFPE tissues[15]. In practice, large-scale and comprehensive transcriptomic profiling of clinical specimens is always required to identify predictive biomarkers or rare cell types. Therefore, the overarching goal is to have an approach that can meet the need for high-throughput, high-sensitivity, and high-coverage snRNA-seq on FFPE tissues.

In this study, we develop snRandom-seq, a droplet-based high-sensitive and full-length snRNA sequencing method for FFPE tissues. In snRandom-seq, we capture total RNAs using random primers for reverse transcription and synthesize the second strand by performing poly(dA) tailing on the first strand cDNAs. cDNAs in a single nucleus are further specifically tagged by our previous microfluidic barcoding platform[16,17] and subsequently amplified and sequenced. Meanwhile, we develop a protocol for isolating single intact nuclei from FFPE tissues by performing deparaffinization, rehydration, and nucleus extraction under mild conditions. Moreover, we design a single-strand DNA-blocking step to avoid the effect of genome contamination. We use a human-mouse mixture sample to validate the performance of snRandom-seq, and the results show a minor doublet rate (0.3%) and a comparable sensitivity with state-of-art high-throughput scRNA-seq technologies. On FFPE mouse tissues, snRandom-seq demonstrates decent results for snRNA sequencing and cell type annotation, where snRandom-seq detects a median of >3000 genes per nucleus for ~20,000 single nuclei and identifies 25 typical cell types (hepatocyte, germ cells, fibroblast, cardiomyocyte, etc.). Moreover, we apply snRandom-seq on a clinical FFPE human liver cancer specimen and reveal an interesting subpopulation of nuclei with high proliferative activity, which might be a potential target for cancer research. In brief, snRandom-seq provides a powerful snRNA platform for laboratory and clinical FFPE specimens and implicates various future applications in biological research and clinical practice.

## Results

### Overview of the droplet-based snRandom-seq method for FFPE tissues

The main workflow of snRandom-seq is shown in Fig. 1. For single nucleus isolation of FFPE tissues, the areas of interest of banked FFPE tissue block were first selected and placed into tubes. Deparaffinization and rehydration were carried out with standard xylene and alcohol wash. Afterward, nuclei were dissociated and permeabilized. For comprehensive and high-throughput single nucleus total RNA-seq, we provided a strategy with a random-primer-based chemistry to capture full-length total RNAs, and an easy-to-operate droplet-based platform to tag single nucleus. Bare single-strand DNAs were blocked in situ by multiple annealing and extension of blocking primers. cDNAs of total

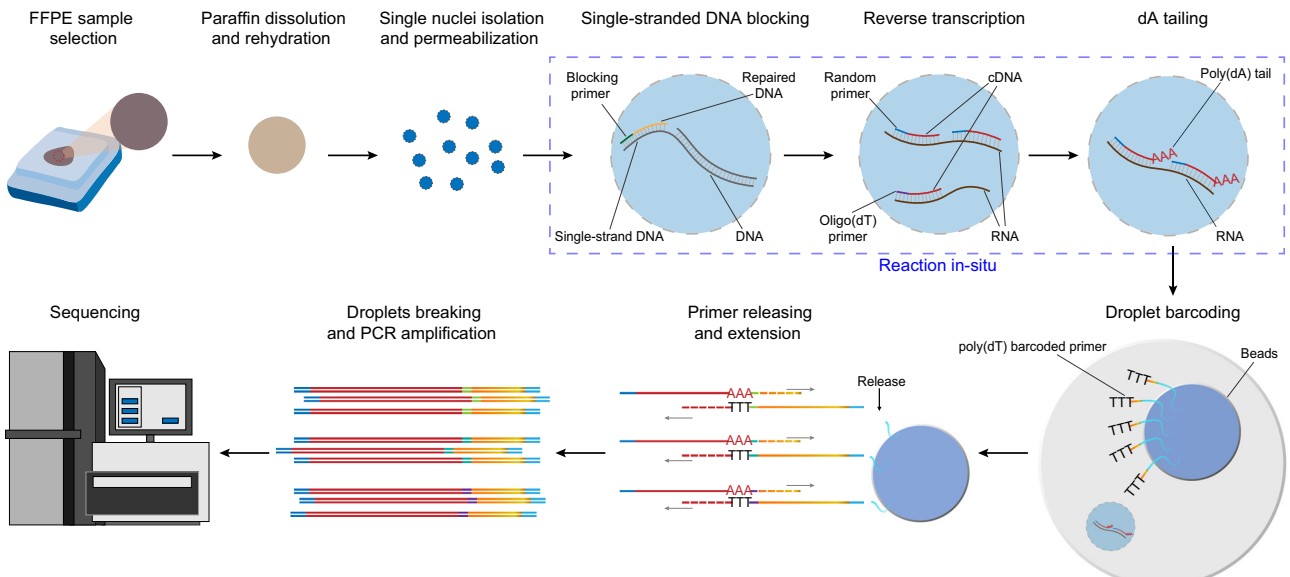

**Fig. 1 | snRandom-seq for FFPE tissues overview.** The workflow of snRandom-seq for FFPE tissues includes FFPE sample selection, paraffin dissolution, single nuclei isolation, and permeabilization, single-strand DNAs blocking, reverse transcription, dA tailing, droplet barcoding, primers releasing and extension, droplets breaking and PCR amplification, and sequencing. Red dashed circle in the FFPE tissue block: the areas of interest. Blue dashed box: the three in situ reactions, including single-strand DNAs blocking, reverse transcription, dA tailing. AAA: dA tail in the 3' of cDNA. TTT: poly(dT) in the poly(dT) barcoded primers. Gray arrows: the direction of extension.

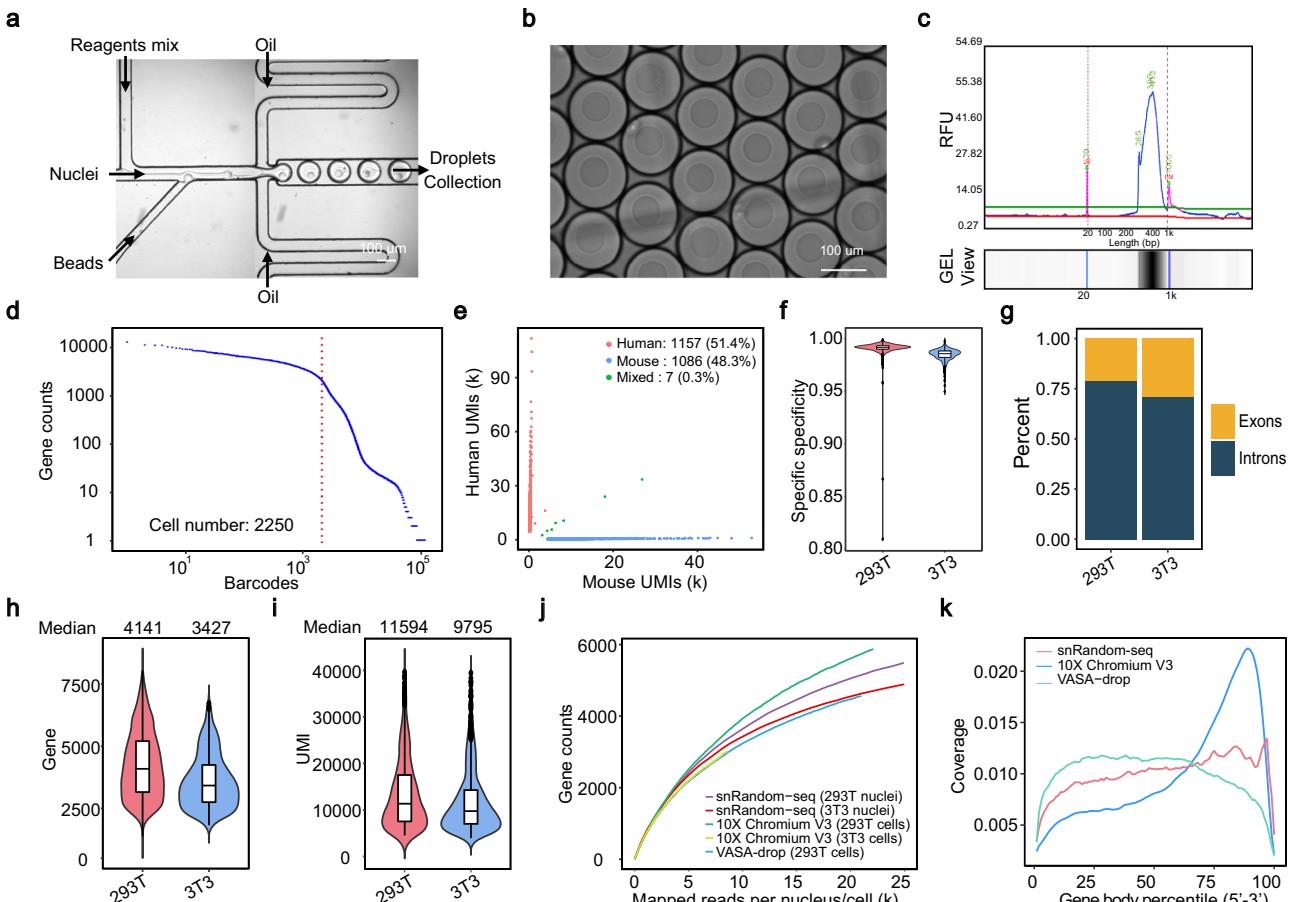

**Fig. 2 | Validation and benchmark of snRandom-seq using a human-mouse mixture sample. a** Microfluidic encapsulation device for barcoding of nuclei. **b** Image of encapsulated droplet containing one bead, one nuclei, and reagents mix. **c** Electropherogram of 293T (human) and 3T3 (mouse) nuclei mixture cDNA library for Qsep100™ DNA Fragment Analyzer. Lower (20 bp) and upper (1 kb) markers were shown. **d** Barcode plot for identification of the barcodes that represent true nuclei (red line). Barcodes of the 293T-3T3 mixed nuclei were ordered from the largest to smallest gene counts. **e** Species-mixing scatter plot showing the single-nuclei capture efficiency and doublet rate of snRandom-seq. **f** Species specificity of UMIs in 293T-3T3 mixture. Identified 293T nuclei: $n = 1157$, identified 3T3 nuclei: $n = 1086$. Median of species specificity of UMIs in 293T was 0.992. Median of species specificity of UMIs in 3T3 was 0.986. **g** Percents of reads mapped to introns and exons. Violin plots and box plots showed the number of genes (**h**) and UMIs (**i**) detected in each 293T and 3T3 nucleus. Filtered 293T nuclei:

$n = 1085$, filtered 3T3 nuclei: $n = 1066$. Data in the box plot corresponded to the first and third quartiles (lower and upper hinges) and median (center). **j** Saturation analysis of three methods. snRandom-seq used 293T and 3T3 nuclei; 10X Chromium Single Cell 3′ Solution V3 used 293 T and 3T3 cells; VASA-seq used 293T cells. **k** Read coverage along the gene body by the three methods. snRandom-seq used 293T nuclei; 10X Chromium Single Cell 3′ Solution V3 used 293T cells; VASA-seq used 293T cells. Data in (**f**, **h**, **i**) were presented as median values. Data in the box plot in (**f**, **h**, **i**) corresponded to the first (lower hinges) quartiles, third quartiles (upper hinges), and median (center). The upper whisker extended from the hinge to the maxima no further than 1.5 * IQR (interquartile range) from the hinge. The lower whisker extended from the hinge to the minima at most 1.5 * IQR of the hinge. IQR is the distance between the first and third quartiles. Source data are provided as a Source Data file.

RNA were converted in situ by multiple annealing of random primers and oligo(dT) primers in reverse transcription. To decrease the doublet rate, we involved a pre-indexing strategy into the reverse transcription step according to the published scifi-RNA-seq[18]. The nuclei were split into different tubes for reverse transcription with pre-indexed random primers, then pooled for the subsequent reaction. Poly(dA) tails were added to the 3′ hydroxyl terminus of the cDNAs in situ by terminal transferase (TdT). We also established a microfluidic platform for high-throughput single nucleus barcoding based on our previous work[16,17]. During the barcoding reaction in droplets, the poly(dT) primers were released from beads by enzymatic cutting[19], and simultaneously, the cDNAs were released from the nucleus by RNA degradation. Then poly(dT) primers bound with the poly(dA) tail on the end of the cDNAs and extended to add a specific barcode to the cDNAs in each droplet. After barcoding, we broke the droplets, amplified the barcoded cDNA, and prepared the next-generation sequencing (NGS) library for paired-end sequencing.

## Validation of snRandom-seq using the human-mouse mixture sample

snRandom-seq utilizes random primers to capture total RNAs in single nuclei (Fig. 1), which differs from the current poly(A)-based and probe-based single-cell RNA-seq methods. Therefore, we performed a standard mixed species experiment with cultured human (293T) and mouse (3T3) cell lines to assess the fidelity of snRandom-seq. Freshly harvested 293T and 3T3 cells were lysed into nuclei and mixed for fixation. The fixed nuclei were used for snRandom-seq (Fig. 1). Before proceeding with microfluidic encapsulation, the nuclei were imaged to confirm single nucleus morphology and counted (Supplementary Fig. 1a). A high-throughput microfluidic platform was established for single cell/nuclei barcoding in snRandom-seq (Fig. 2a, Supplementary Fig. 1b). For barcode beads synthesis, the hydrogel bead generation device and the cell encapsulation device were designed and fabricated as previously described[20] (Supplementary Fig. 2a). Hydrogel beads of 40 μm diameter were precisely produced (Supplementary Fig. 2b).

Three rounds of split-and-pool-based ligation were performed on these hydrogel beads for DNA barcode synthesis (Supplementary Fig. 2c, Supplementary Table 2). The high reaction efficiency of each ligation step was reflected by the sharp peak in the electropherogram of released barcode primers (Supplementary Fig. 2d). Nucleus, barcode bead, and reagents mix were co-compartmentalized in water-in-oil emulsions using the microfluidic platform (Fig. 2a) and each individual nuclei were encapsulated into a droplet with a barcode bead (Fig. 2b).

After barcoding and amplification, the fragment size of the cDNA library of the human-mouse mixture peaked between 300 and 800 bps (Fig. 2c), which is not needed to fragment but is just suitable for NGS. After data processing, we identified 2250 high-quality unique nucleus barcodes by the significant steep slope in the barcode-gene rank plot (Fig. 2d), which suggests a clear separation of true nuclei from background noise. The nuclei capture rate was 42.2% and the percentage of reads mapped to the true nuclei was 76%. We counted the ratio of reads mapped to both human and mouse genomes in every single nucleus and found that pre-indexed primers markedly decreased the doublet rate (from 2.9% to 0.3%) (Fig. 2e, Supplementary 1c). The doublet rate of snRandom-seq is significantly lower than that of other droplet-based sc/snRNA-seq methods (sNucDrop-seq: ~2.6%, VASA-drop: 3.1%). Consistently, very high species specificity of UMI (99%) was observed (Fig. 2f), suggesting that snRandom-seq produced high-fidelity single nucleus libraries. The percentage of the reads mapped to exon or intron of identified human and mouse nuclei was calculated, and the results showed that the reads mapped to intron were three times of the reads mapped to exon (Fig. 2g). Additionally, many long non-coding RNAs (lncRNAs) and short non-coding RNAs, including small nucleolar RNA (snoRNA), small nuclear RNA (snRNA) and microRNA (miRNA), were detected (Supplementary Fig. 1d). Those results suggested that snRandom-seq captured full-length transcripts comprehensively.

Gene and UMI count distribution showed that snRandom-seq captured a median of 4141 genes and 11,594 UMIs in single 293T nucleus by sequencing average ~29k reads per 293T nucleus (Fig. 2h), and 3427 genes and 9795 UMIs in single 3T3 nucleus by ~25k reads per 3T3 nucleus (Fig. 2i). The results indicated that snRandom-seq is more sensitive than other two reported droplet-based high-throughput snRNA-seq methods (DroNc-seq[21]: average 3295 genes and 4643 UMIs with ~160k reads per nucleus for 5636 3T3 nuclei; sNucDrop-seq[22]: average 2665 genes and 5195 UMIs with ~23k reads per nucleus for 1984 3T3 nuclei) (Supplementary Fig. 1e). Saturation analysis showed that the number of genes detected in snRandom-seq had not yet reached saturation point by 60k uniquely aligned reads per 3T3 and 293T nucleus (Fig. 2j). We also compared our snRNA-seq data to the widely used high-throughput 10X Chromium Single Cell 3′ Solution V3[23] and the latest reported high-throughput VASA-drop[10] for scRNA-seq. At a low sequencing depth (<10k), the sensitivity of snRandom-seq in 3T3 and 293T nuclei is comparable with 10X Chromium Single Cell 3′ Solution V3 in 3T3 and 293T cells, as well as VASA-drop in 293T cells (Fig. 2j). Unlike poly(A)-based 10X Chromium Single Cell 3′ Solution V3 with obvious 3′-end bias, both snRandom-seq and VASA-drop displayed no obvious 3′- or 5′-end bias across the gene body (Fig. 2k). As expected, snRandom-seq had a slight bias toward the 3′-end due to the extra addition of oligo(dT) primer in reverse transcription (Fig. 2k).

## Performance of snRandom-seq in the FFPE tissues

For FFPE tissues, digestion with Proteinase K could isolate cleaner single nuclei than with collagenase (Supplementary Fig. 3a). With an optimized procedure (Fig. 1), single intact nuclei were efficiently isolated from multiple FFPE mouse tissues and a 2-year-old archived clinical FFPE sample of human liver cancer (Fig. 3a, Supplementary Fig. 4a), and the nuclei morphology and size distribution were comparable between FFPE and fresh samples (Supplementary Fig. 4b).

In our pilot FFPE snRNA sequencing experiment, little uniquely aligned reads were mapped to exons, with many reads mapped to intergenic regions due to genome contamination (Fig. 3b). Considering that the double-helix of DNA in FFPE tissues is liable to be disrupted after suffering chemical modification, a single-strand DNAs blocking step was added to the initial procedure of snRandom-seq (Fig. 1, box). The bare single-strand DNAs in the isolated FFPE single nucleus were blocked in situ by multiple annealing and extension of blocking primers on single-strand DNAs of genome. After DNA blocking, the percentage of intergenic regions was dramatically reduced (Fig. 3b). The mapping region distribution was comparable among DNA-blocked FFPE sample, fresh sample, and snFFPE-seq (10X Chromium Single Cell 3′ Solution V3), further supporting the high quality of the snRandom-seq data (Fig. 3b). By integrating the above procedures, high-quality cDNA libraries were generated by snRandom-seq from multiple FFPE tissues (Fig. 3c, Supplementary Fig. 4c, d). The fragment size of cDNA libraries from FFPE and fresh samples both peaked between 300 and 800 bps (Fig. 3c, Supplementary Fig. 4e).

To determine whether snRandom-seq can generate enough information from FFPE tissues as fresh samples, we collected both FFPE and fresh samples from the same mouse tissues and compared their RNA profiles using snRandom-seq (Fig. 3d). The RNA quality of FFPE and fresh samples were evaluated firstly by the RNA fragments distribution and DV200. As expected, the RNA quality of the FFPE sample was relatively poorer than that of the fresh sample (Supplementary Fig. 5a), suggesting that the RNA in the FFPE sample was degraded. The merged genome browser tracks of snRandom-seq results showed that the reads coverage areas of FFPE and fresh samples were similar (Supplementary Fig. 6a–g). Consistently, the total RNA profiles of FFPE and fresh samples by snRandom-seq displayed a good correlation (Pearson R: ~0.9, $p < 2.2e\text{-}16$; Fig. 3e, Supplementary Fig. 7a, b). Meanwhile, to prove the repeatability of our method, the same FFPE sample was sequenced independently with snRandom-seq (Fig. 3d), and a high correlation (Pearson R ~ 0.92, $p < 2.2e\text{-}16$) of gene expression profiles across these two batches was also seen (Fig. 3f). These results showed that snRandom-seq performed well in both fresh and FFPE samples.

We next compared our FFPE results with other reported FFPE snRNA-seq results. After data processing, thousands of true nuclei in these FFPE tissues were successfully identified from the snRandom-seq data (Fig. 3h, Supplementary Fig. 8a). snRandom-seq identified a broad spectrum of RNA biotypes in the FFPE sample (Fig. 3g), with about eight times as many lncRNAs as snFFPE-Seq, and snoRNA, scaRNAs, and miRNA were only detected in snRandom-seq (Supplementary Fig. 8b). The medians of genes detected per nuclei in unsaturated snRandom-seq datasets were all over 3000, significantly higher than that in other two reported high-throughput snRNA-seq methods for FFPE samples (snFFPE-Seq 10X Chromium Single Cell 3′ Solution V3: 276 genes/nucleus; snPATHO-Seq: 1850 genes/nucleus) (Fig. 3h), as well as the medians of UMIs (Supplementary Fig. 8c). Our data still has not yet to reach saturation point even sequencing ~300k mapped reads per nuclei and detecting ~10,000 genes (Fig. 3i).

We further compared the RNA coverage of snRandom-seq with the other two FFPE snRNA-seq methods. In the plot of average reads distribution on gene body, snFFPE-Seq using oligo(dT) primers showed a distinct 3′-end bias and 10X Chromium Fixed RNA Profiling using the same probe-base technology of snPATHO-seq showed a mild 5′-end bias (Fig. 3j). However, homogeneous distribution across gene body was observed in snRandom-seq data for the FFPE tissue (Fig. 3j), suggesting that random primers were evenly bound on transcripts and the extra oligo(dT) primers in snRandom-seq were invalid for FFPE sample. For RNA coverage at the level of single nucleus, snRandom-seq showed much higher coverage than that of snFFPE-seq or 10X Chromium Fixed RNA Profiling (Fig. 3k). For RNA coverage at the level of single gene, reads distribution along three selected genes (*C1S*, *EMG1*,

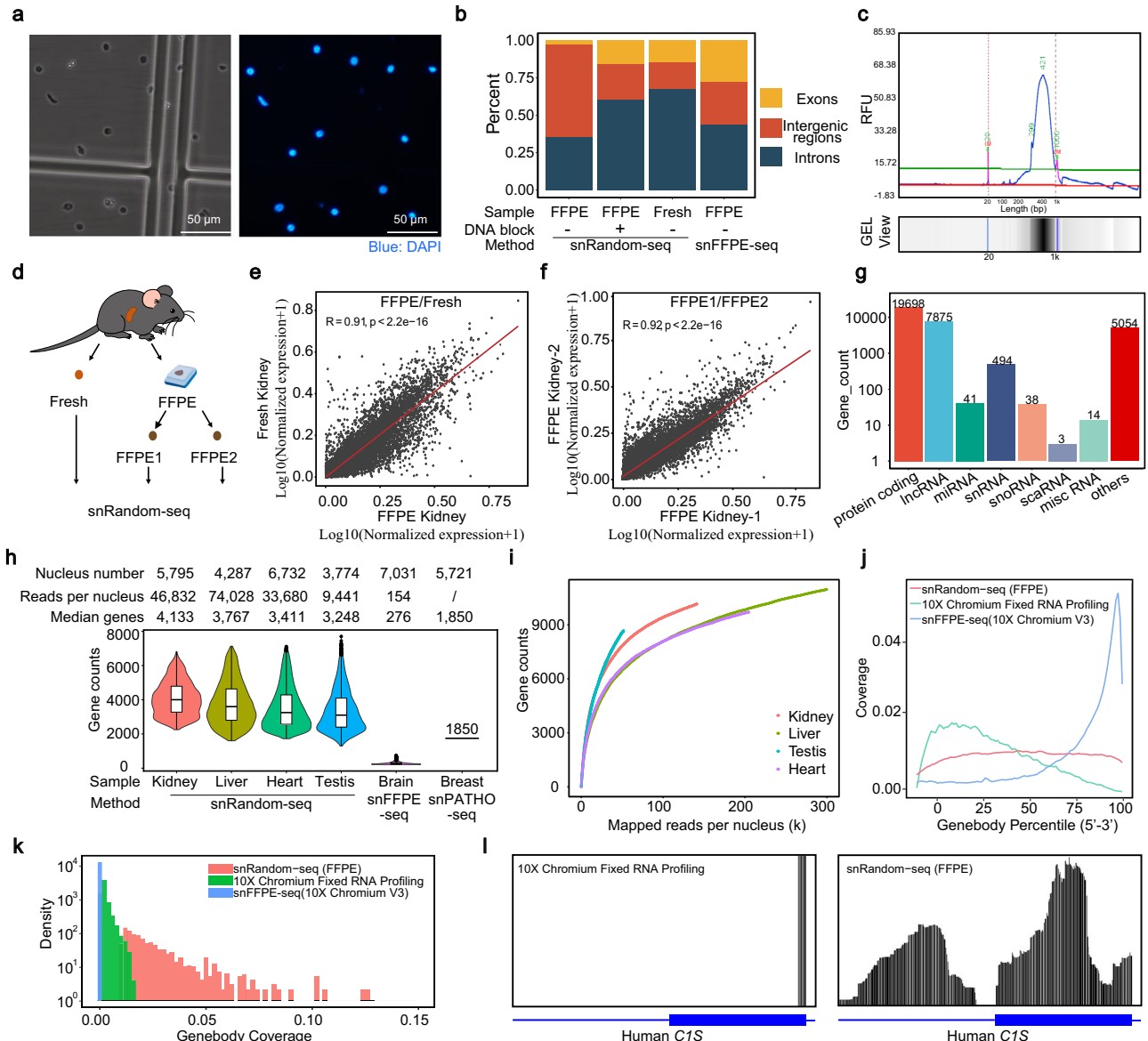

**Fig. 3 | Comparison of snRandom-seq with other two FFPE snRNA-seq methods.**
**a** Image of single nuclei before droplet barcoding and staining by DAPI. Scale bar, 50 μm. **b** Percentage of reads mapped to different genomic regions under different conditions. **c**, Electropherogram of FFPE mouse kidney cDNA library for Qsep100™ DNA Fragment Analyzer. Lower marker: 20 bp; upper marker: 1k bp. **d** Overview of FFPE/fresh comparison and technical replication experiment. The Pearson's correlation coefficient (*R*) of the normalized gene expressions between FFPE/fresh samples (**e**) and technical replication samples (FFPE1, FFPE2) (**f**). Each dot represents the average expression level of a gene. The red line indicates the linear regression line. *p* value (*p*) was computed from two-sided permutation test. **g** Counts of different RNA biotypes detected in FFPE sample. **h** Gene detection comparison of mouse tissues (heart, kidney, testis, and liver) and human liver using snRandom-seq with mouse brain by snFFPE-seq[15] and breast by snPATHO-seq[14].

Kidney nuclei: *n* = 5795, liver nuclei: *n* = 4287, heart nuclei: *n* = 6732, testis nuclei: *n* = 3774, brain nuclei: *n* = 7031, breast nuclei: *n* = 5721. Data were presented as median values. Data in the box plot corresponded to the first (lower hinges) quartiles, third quartiles (upper hinges), and median (center). The upper whisker extended from the hinge to the maxima no further than 1.5 * IQR from the hinge. The lower whisker extended from the hinge to the minima at most 1.5 * IQR of the hinge. **i** Saturation analysis of snRandom-seq based on the FFPE mouse tissues. **j** Reads distribution along the gene body by three different snRNA-seq methods (snRandom-seq, snFFPE-seq[15], and 10X Chromium Fixed RNA Profiling). **k** Histogram showing the gene body coverage percents datasets generated by the three methods. **l** Representative raw reads aligned to human gene *C1S* in snRandom-seq and 10X Chromium Fixed RNA Profiling. Source data are provided as a Source Data file.

*KLRG1*) indicated the critical difference between probe-based technology and the random primer-based strategy (Fig. 3l, Supplementary Fig. 8d). Mapped reads by 10X Chromium Fixed RNA Profiling were limited to the probe-target regions (<100 bp). In contrast, the mapped reads by snRandom-seq were evenly distributed in both exonic and intronic regions. These results suggested that snRandom-seq for FFPE tissues can capture a significant amount of high-quality RNA and extract much more transcriptomic information than the state-of-art platforms.

## snRandom-seq revealed cell heterogeneity in FFPE mouse tissues

We next compared the cell types identified in FFPE and fresh samples by snRandom-seq. Unsupervised clustering of the above filtered high-quality single kidney nucleus profile revealed over ten distinct clusters. All clusters could be further annotated based on classical known cell-type markers[24,25] (Fig. 4a, b, Supplementary Fig. 9a). Gene expressions of classical known cell-type marker genes[22], such as *Nphs1* for podocytes, *Pecam1* for endothelial cells, and *Pdgfrb* for mesangial-like cells,

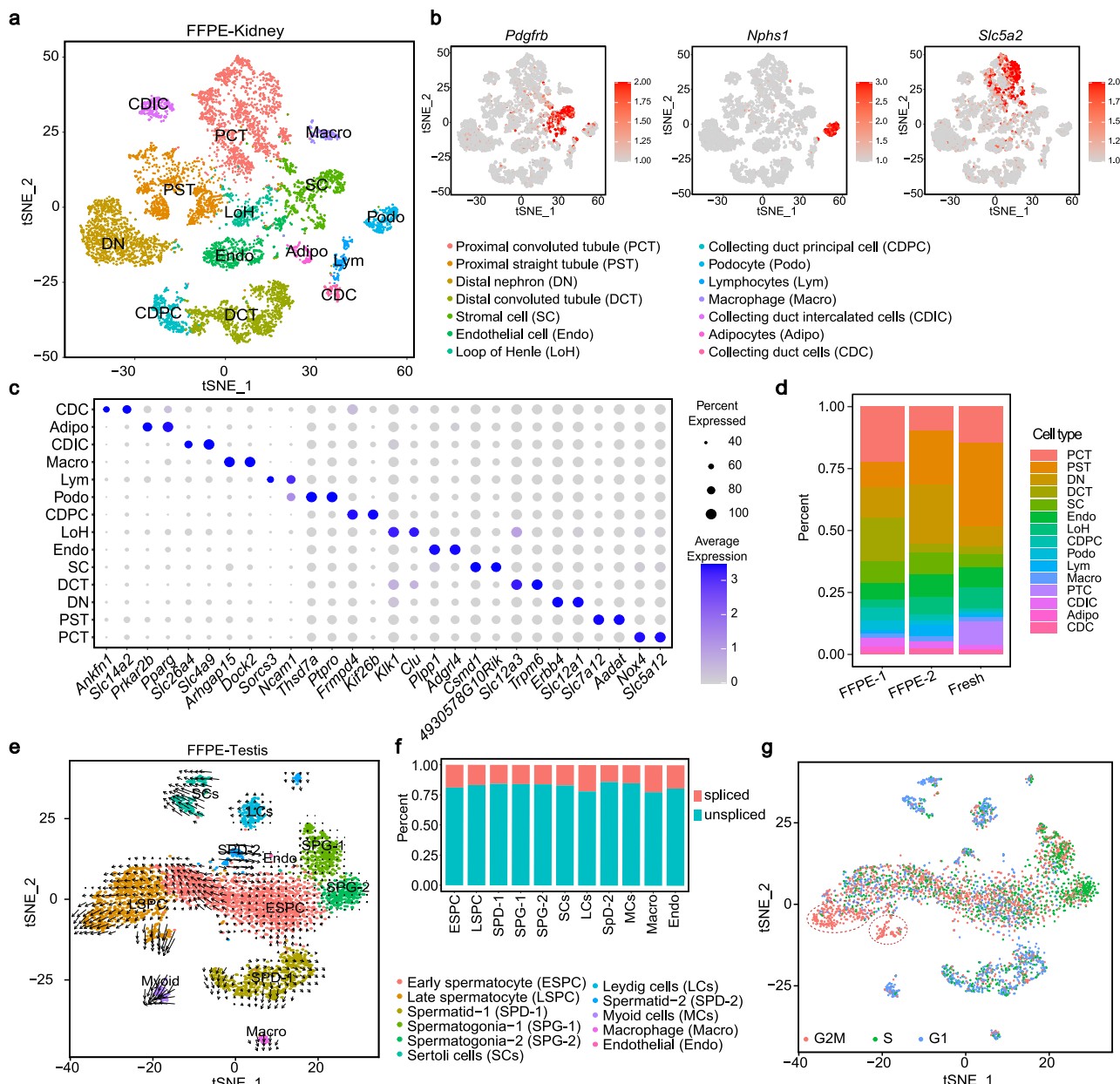

**Fig. 4 | Cell heterogeneity revealed in FFPE mouse tissues by snRandom-seq. a** *t*-SNE analysis of nuclei isolated from FFPE mouse kidney sample by snRandom-seq based on their gene expressions and colored by identified cell types. Fourteen Cell types identified were colored and shown below. **b** Expression of selected three cell-type markers in single nuclei in the *t*-SNE maps of FFPE mouse kidney. Gene expression levels are indicated by shades of red. **c** Dot plot of the average expressions of top two markers in each of the 14 cell types. **d** Proportion of annotated cell types of FFPE1, FFPE2, and fresh samples by snRandom-seq. **e** *t*-SNE and RNA velocity analysis of snRNAs from FFPE mouse testis by snRandom-seq. Velocity is shown as black arrows in different cell types by separate colors. The black arrows indicate RNA maturation trajectory. **f** Percents of spliced and unspliced transcripts in different cell types. **g** Cell cycle analysis of FFPE mouse testis by snRandom-seq. Points in *t*-SNE were colored by identified cell cycle phases (G1, G2M, or S). Red dashed circle: two subpopulations of late spermatocytes at the G2M phase with active transcriptional activity. Source data are provided as a Source Data file.

were reliably mapped on the corresponding clusters (Fig. 4b). The mammalian renal tubule in the kidney contains at least 16 distinct epithelial cell types[26]. Here we identified most of the recommended terms for renal tubule epithelial cell types in FFPE mouse kidney samples by snRandom-seq, including proximal convoluted tubule, proximal straight tubule, distal nephron, distal convoluted tubule, loop of Henle, collecting duct principal cells, podocytes, proximal tubular cells, collecting duct intercalated cells, and collecting duct cells (Fig. 4a). Besides the known top markers of cell types, such as *Slc14a2* for collecting duct cells, we also discovered several potential markers for these cell types (Fig. 4c). By merging the snRandom-seq

data of the FFPE samples and fresh sample, as well as the other batch of FFPE samples, we obtained a robust cell clustering by *t*-SNE (*t*-distributed stochastic neighbor embedding) (Supplementary Fig. 9b). Most cell types were identified in the three snRandom-seq datasets (Fig. 4d, Supplementary Fig. 9c). As expected, there are some differences in the proportion of cell types of the FFPE and fresh samples (such as PTC), which might be caused by the sampling error and different nuclei extraction methods for FFPE and fresh samples.

We further added more FFPE mouse tissues to demonstrate the biological utility of snRandom-seq data. In total, we sequenced and analyzed 19,258 single nuclei from four FFPE mouse tissues (heart,

kidney, testis, and liver) using snRandom-seq and identified a total of 25 cell types (such as hepatocyte, germ cells, fibroblast, cardiomyocyte, etc.). (Supplementary Fig. 10a, b). An underrepresentation of immune cells could be seen, which is consistent with previous findings about cell type composition by single-nucleus RNA-seq libraries[27].

The large proportion of intronic sequences detected in FFPE samples (Fig. 3b) suggested that snRandom-seq data would be more suitable for RNA velocity analysis by distinguishing newly transcribed RNAs (unspliced) from mature RNAs (spliced)[28]. Next, we applied snRandom-seq to FFPE mouse testis, where spermatogenesis is an excellent model for studying cell dynamics. Consistently with other studies on fresh testis by scRNA-seq[29,30], t-SNE arranged germ cells at transitionary stages (mainly early spermatocyte and late spermatocyte) to be in continuous succession. In contrast, undifferentiated spermatogonia and mature spermatids are in clusters (Fig. 4e). The velocities computed by detected nascent transcripts were visualized on the t-SNE plot, revealing distinct velocity vector directions in different cell types, especially in the cells located at the left of early and late spermatocytes (Fig. 4e, f). Combined with cell cycle states analysis based on gene expression, the RNA velocity revealed an obvious cell maturation trajectory on two subpopulations of late spermatocytes at the G2M phase with active transcriptional activity (Fig. 4g).

### snRandom-seq discovered a proliferative subpopulation in the FFPE clinical human specimen

Finally, we applied snRandom-seq on an about two-year-old clinical FFPE specimen of human macrotrabecular-massive (MTM) hepatocellular carcinoma (HCC) subtype (Fig. 5a). We selected an interested tumorous area on the paraffin block according to the histopathological examinations (Fig. 5b) and performed snRandom-seq. snRandom-seq identified 5914 true nuclei and detected a median of 3220 genes and a median of 8182 UMIs per nucleus in this clinical FFPE specimen (Supplementary Fig. 11a, Fig. 5b). As sequencing depth increases, snRandom-seq detected about 8000 genes at saturation (Supplementary Fig. 11b). A broad spectrum of RNA biotypes including lncRNAs, snRNAs, miscRNAs, miRNAs, and snoRNAs was detected from the sample (Supplementary Fig. 11c). Unsupervised clustering of the human liver single nucleus revealed several distinct clusters. The main cell types of human liver could be identified from the human specimen based on the known cell-type markers[31], including hepatocyte (APOA1), kupffer cells (CD163), T cells (CD3E), fibroblast (PDGFB), plasma cells (FCRL5) (Fig. 5d, Supplementary Fig. 11d). Notably, a subcluster of hepatocytes (hepatocyte-2) was separated from the main hepatocyte population, with high expression of the proliferative marker MKI67 and the other two markers (ASPM and TOP2A), which were reported to be related to HCC progression[32,33]. (Fig. 5e). Meanwhile, cell cycle analysis of these snRNAs revealed that most cells in the hepatocyte-2 cluster were in phase G2M (Fig. 5f), suggesting that the hepatocyte-2 cluster might be a group of dividing tumor cells. After further investigating the cell communication among the clusters (Fig. 5g), we found that hepatocyte-1 and hepatocyte-2 displayed different outcoming and incoming signaling patterns (Fig. 5h). Hepatocyte-2 mainly receives signals from plasma cells through the BMP signaling pathway (Supplementary Fig. 12a), which is reported to be correlated with tumor progression in HCC[34,35]. Ligand–receptor pair analysis found that plasma cells preferentially sent signals to hepatocyte-2 by BMP6-(ACVR1 + ACVR2A) and the communication between plasma cells and hepatocyte-2 has specific ligand-receptor pairs, including BMP6-(BMPR1B + BMPR2), BMP6-(BMPR1B + ACVR2B), BMP6-(BMPR1B + ACVR2A), BMP6-(BMPR1A + ACVR2A), and BMP6-(ACVR1 + ACVR2A) (Fig. 5i). The gene expression also showed that BMPR1B and ACVR2A have specific expressions in hepatocyte-2 (Supplementary Fig. 12b). Taken together, snRandom-seq discovered a proliferative and activated subpopulation of hepatocytes from a clinical FFPE specimen, which provides a valuable clue for additional study in future.

snRandom-seq was also performed on an FFPE specimen of human normal HCC subtype (Supplementary Fig. 13a). Based on the snRandom-seq data, sufficient gene count and UMI count were detected, and main cell clusters of the liver were identified (Supplementary Fig. 13b, c). Previous studies have indicated that lncRNAs exhibit tissue-specific expression[36,37], which is always ignored in routine single-cell RNA-seq analysis due to their low expression. We found that hepatocyte clusters of normal HCC subtype had a markable expression of lncRNAs, including LINC02476 and LINC01151 in hepatocyte-2, LINC00540, LINC02307, and LINC02109 in hepatocyte-3, LINC02384 in hepatocyte-4 (Supplementary Fig. 13d). It has been reported that LINC02476 promotes the malignant phenotype of HCC by sponging miR-497 and increasing HMGA2 expression[38], and LINC00540 influences human HCC progression and metastasis via the NKD2-dependent Wnt/β-Catenin Pathway[39]. These results suggested that hepatocyte-2 (expressed LINC02476) and hepatocyte-3 (expressed LINC00540) of normal HCC subtype might exhibit different pathogenesis. Taken together, snRandom-seq with the advantages of full-length transcripts coverage shows promise in lncRNA analysis in cancer biology.

We further performed an application of snRandom-seq on a matched pair of initial and relapsed FFPE clinical specimens from the same colorectal cancer liver metastasis (CRLM) patient. snRandom-seq detected medians of ~1000 gene counts and ~2000 UMI counts in both initial and relapsed FFPE specimens (Supplementary Fig. 14a). The cells from the initial and relapsed FFPE specimens were comprehensively integrated, and the major cell types (hepatocytes, cancer cells, T cells, fibroblasts, myeloid cells, endothelial cells, stellate cells, macrophages, cholangiocytes, B/plasma cells) were identified in both samples (Supplementary Fig. 14b, c). We observed that the proportion of T cells was higher in the relapsed FFPE sample (Supplementary Fig. 14d), suggesting a more active antitumor immune response in the relapsed sample. Consistently, the proportions of the dominating cancer clusters (cancer cells-1, −2, and −3) were decreased in the relapsed sample (Supplementary Fig. 14d). However, the proportion of cancer cells-4 was increased in the relapsed sample (Supplementary Fig. 14d). We further found that the genes encoding lipids composition regulator (SCD) and proteins binding lipids (APOA2, APOC3, and APOA1) displayed high expression levels in cancer cell-4 cluster in the relapsed sample (Supplementary Fig. 14e), suggesting an enhanced lipid metabolism in the cancer cells subcluster of the relapsed CRLM.

## Discussion

We developed a droplet-based high-throughput snRNA-seq method for archived FFPE tissues by using random primer to capture full-length total RNAs from single nuclei sensitively and comprehensively, which therefore provides a critical advance to profile single nuclei transcriptome from FFPE tissues or other types of low-quality biological samples. It is worth noting that snRandom-seq uses routine molecular biology procedures and mature microfluidic droplet barcoding platform that is similar with the current popular 10X Genomics platform. Therefore, snRandom-seq is easy to operate and commercialize for large-scale applications.

Molecular biological application of FFPE tissues has always been challenging due to the chemical cross-linked and low-quality RNA. Although single nuclei could be isolated from FFPE tissues and the RNA crosslink could be reversed by heat and protease digestion, the popular oligo(dT)-based RNA capture strategy is not efficient with these low-quality samples as demonstrated by the snFFPE-seq with 10X Chromium Single Cell 3′ Solution V3 platform[15], as well as the invalid oligo(dT) primer in our snRandom-seq. snPATHO-Seq[14], which adopted 10X Genomics probe-based technology for FFPE nucleus, can reflect the gene signatures for genes of interest, whereas it only targets

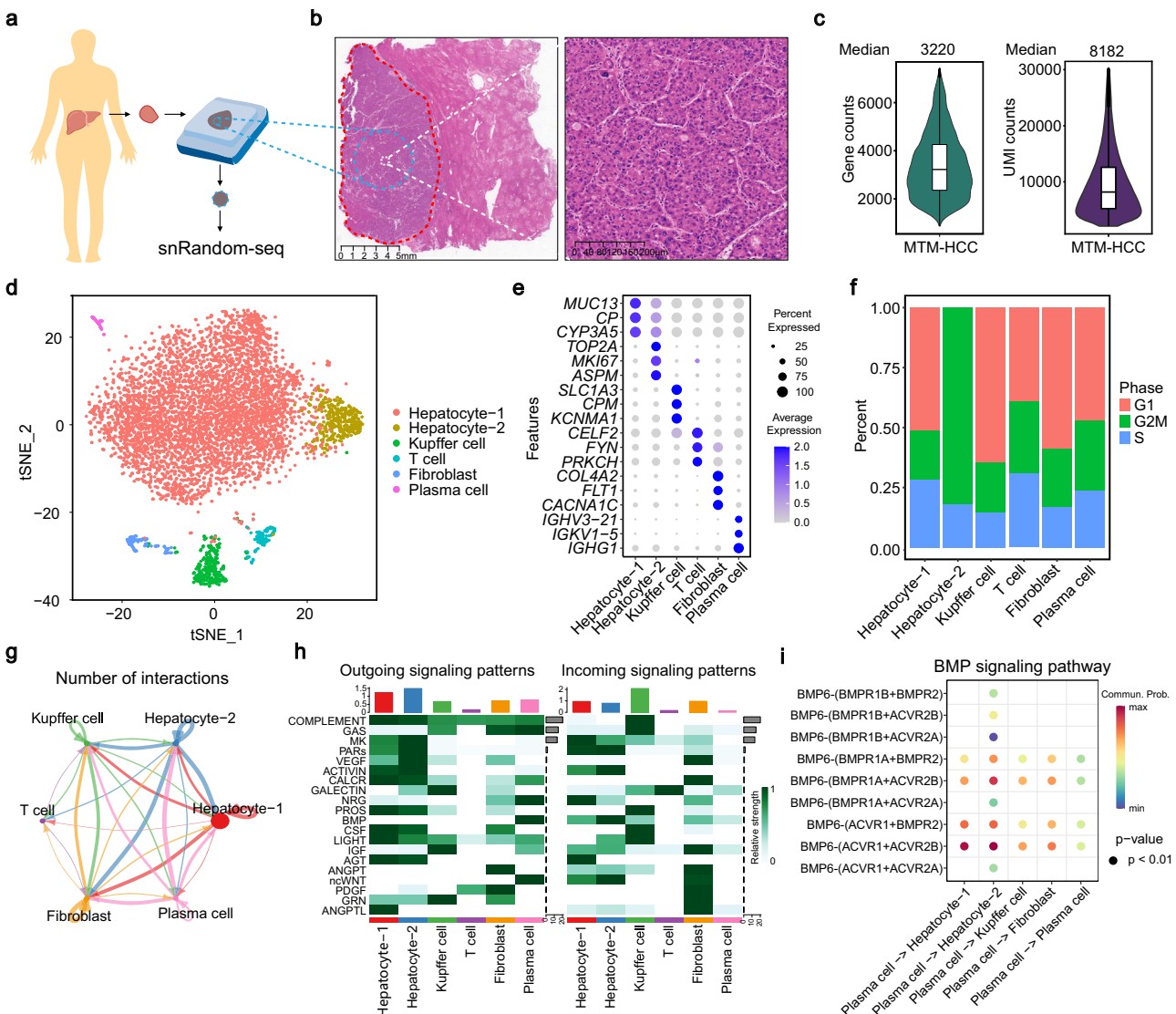

**Fig. 5 | snRandom-seq discovered a proliferative subpopulation in the clinical FFPE human specimen. a** Experimental overview of clinical FFPE sample of human macrotrabecular-massive hepatocellular carcinoma (MTM-HCC) subtype for snRandom-seq. **b** Histological appearance of MTM-HCC at low magnification (left, Scale bar, 5 mm.) and high magnification (right, Scale bar, 200 µm). Red dashed circle: tumorous area. Blue dashed circle and line: sampling area. White dashed circle and line: magnified area. **c** Violin plots and box plots showing the number of genes and UMIs detected in FFPE clinical human sample by snRandom-seq. MTM-HCC nuclei: *n* = 5914. Data was presented as median values. Data in the box plot corresponded to the first and third quartiles (lower and upper hinges) and median (center). Data in the box plot corresponded to the first (lower hinges) quartiles, third quartiles (upper hinges), and median (center). The upper whisker extended from the hinge to the maxima no further than 1.5 * IQR from the hinge. The lower whisker

extended from the hinge to the minima at most 1.5 * IQR of the hinge. **d** *t*-SNE map of nuclei isolated from the FFPE sample based on their gene expressions. Six cell types, including two hepatocellular subtypes, were annotated and shown. **e** Top three markers of each of the six cell types. **f** Percentages of the nuclei in phase G1, G2M, or S in the six cell types. **g** The total number of interactions among different cell populations. Circle sizes represented the number of cells in each cell group and edge width represented the communication probability. **h** The heatmap showing the outgoing signaling patterns (left) and incoming signaling patterns (right) of each cell cluster. **i** Bubble diagrams showing the communication probability and statistical significance of receptor-ligand pairs in BMP signaling network. Dot color represented communication probabilities and dot size represented computed *p* values. Empty spaces mean that the communication probability was zero. *p*-values were computed from one-sided permutation test. Source data are provided as a Source Data file.

---

very small part of transcriptomes. While using the random-primer-based approach, we are able to perform unbiased single-nucleus transcription profiling efficiently on the FFPE samples. We also envisage an integration of snRandom-seq chemistry with the spatial barcoding technology[3,40], which enables high-sensitivity and comprehensive spatial gene expression analysis in FFPE tissues, pairing with routine histology. Microbe is another type of challenging sample for scRNA-seq, whose mRNAs content is very low and 3′-end poly(A) tails are usually lacked[41]. We are attempting to modify snRandom-seq and expand its application in high-throughput and high-sensitivity single microbe RNA-seq.

Compared with the state-of-art high-throughput snRNA-Seq methods on the FFPE samples[14,15], snRandom-seq outperforms these methods from various perspectives, supported by the decent performances on cell type identification, differential expression analysis, and the cell cycle phases analysis. High-quality and high-sensitivity snRNA-seq data from clinical FFPE specimens by snRandom-seq allows to reveal of cell-type-specific target genes or identify rare subpopulations of precision diagnosis and treatment to human disease. In addition, random primers, allow snRandom-Seq to cover more gene body regions, which enables the detection of a large amount of non-coding RNAs and the further utility of nascent transcripts in RNA

velocity analysis. We have utilized nascent transcripts for RNA velocity analysis and revealed obvious cell maturation trajectory on two specific subpopulations in this study. Moreover, these full-length total snRNA-seq datasets allow a comprehensive analysis of copy number variation (CNV), alternative splicing, and mutations at single-cell/ nucleus level.

In conclusion, the simple experimental protocols and comprehensive transcriptomic information from the FFPE tissues described in this study are expected to enable snRandom-seq to large-scale applications in basic and clinical researches in the future.

## Methods

### Ethical statement

All the procedures involving mice in this study were approved by Zhejiang University Animal Care and Use Committee (approval numbers: ZJU20170466). The collection of human samples and research conducted in this study was approved by the Research Ethics Committee of the First Affiliated Hospital, Zhejiang University School of Medicine (approval numbers: IIT20220893A). Clinical information was collected after written informed consent. This study is compliant with the Guidance of the Ministry of Science and Technology (MOST) for the Review and Approval of Human Genetic Resources (approval numbers: 2023BAT0303).

### Experimental model

HEK293T cells (Cat. CL-0005) and 3T3 cells (Cat. CL-0006) were ordered from company (Procell Life Science&Technology). Male wildtype C57BL6/J mice (6–8 weeks of age) were ordered from Shanghai SLAC Laboratory Animal. Only male mice were used in the study. Sex was determined based on similar studies in this field. Mice were single-housed under standard laboratory conditions, including a 12 h light/dark cycle, temperatures of 18–23 °C with 40–60% humidity, with free access to mouse diet and water. All experiments conformed to the relevant regulatory standards at Zhejiang University Laboratory Animal Center. FFPE samples of mouse tissues were prepared by Core Facilities, Zhejiang University School of Medicine. FFPE tissues of clinical human cancers were provided by the First Affiliated Hospital, Zhejiang University School of Medicine. The FFPE tissues of macrotrabecular-massive hepatocellular carcinoma and normal HCC were collected from two male Chinese patients (age 43 and 45, respectively) by surgical resection. The matched pair of initial and relapsed FFPE clinical specimens were collected from the same male Chinese patient (initial age 62, relapsed age 64) with colorectal cancer liver metastasis. Clinical information was collected after writing informed consent.

### Species mixture experiment

HEK293T cells and 3T3 cells were maintained in Dulbecco's Modified Eagle's medium (DMEM, Gibco, Cat #11965092), supplemented with 10% (v/v) heat-inactivated fetal bovine serum (FBS, Gibco, Cat # 26010074), and cultured at 37 °C in a 5% $CO_2$ incubator (Thermo Heracell 240i). Both cells were passaged every 2 days. For the species mixture experiment, HEK293T cells and 3T3 cells were harvested and washed three times in PBS by centrifuging at 4 °C, 600 g for 3 min. Cells were lysed by pre-cold nuclei lysis buffer (1X PBS with 0.1% Nonidet P-40 (NP-40, Aladdin, Cat # N274254) and 1 U/μL RNase Inhibitor (Invitrogen, Cat # N8080119)) incubating at 4 °C for 5 min. Then, the fresh nuclei were washed three times and fixed by adding 1 mL of 4% paraformaldehyde (PFA, Aladdin, Cat # P395744) in PBS and incubating at room temperature for 15 min. Next, the PFA was discarded by centrifuging at 600 g for 3 min, and nuclei were washed three times with 1 mL of pre-cold wash buffer (1X PBS with 1 U/μL RNase Inhibitor). Nuclei were permeabilized by adding 500 μL of 0.1% Triton X-100 (Aladdin, Cat # T109027) diluted in pre-cold wash buffer, and incubated at 4 °C for 5 min. Then, 1 mL of wash buffer was added

directly to the nuclei, and the nuclei were washed three times with 1 mL of pre-cold wash buffer. HEK293T nuclei and 3T3 nuclei were counted respectively and equally mixed. Then, the mixture was processed to single nuclei RNA-seq according to the following snRandom-seq protocol.

### Single nuclei isolation from FFPE samples

FFPE samples were cut from the paraffin block and were washed twice with 1 mL Xylene (Aladdin, Cat # X112054) for 5 min at room temperature to remove the paraffin. The samples were gently redehydrated by immersing the samples in a graded series of ethanol solutions (Aladdin, Cat # E130059), starting with pure 100% ethanol and ending with 30% ethanol. The samples were then washed twice with pre-cold wash buffer and homogenized with Dounce homogenizer (Bellco Glass, Inc. Dounce Homogenizer 2 mL, Cat # 50-194-5204) with the presence of pre-cold lysis buffer (1X PBS buffer, 0.1% Triton X-100, 1 U/μL RNase Inhibitor) on ice. After homogenization, an additional 1 mL of lysis buffer was used to rinse the douncer, and 100 μL of 10 mg/mL proteinase K (Sangon Biotech, Cat # A610451) was added into the lysis buffer, incubating at 37 °C for 5 min. Then, the isolated nuclei were filtered through a 20-μm cell strainer (pluriSelect, Cat # 43-10020-40) and washed twice with wash buffer. An aliquot of nuclei was stained with DAPI (4′,6-diamidino-2-phenylindole) staining solution (Abcam, Cat # ab228549), loaded on a hemocytometer and observed under an inverted fluorescence microscope (Nikon Eclipse, Ts2-FL). The qualified single nuclei were processed to single nuclei RNA-seq according to the following snRandom-seq protocol. A detailed protocol, including the volume of the lysis buffer and permeabilization buffer, reaction systems, and reaction programs, was provided in the Supplementary Information file (Supplementary Note 1: snRandom-seq protocol 1.0).

### FFPE and fresh samples comparison experiment

Fresh samples of adult mouse tissues were harvested from the same mouse for FFPE samples. Fresh samples were washed twice with pre-cold wash buffer, cut into pieces, and homogenized with Dounce homogenizer with pre-cold lysis buffer on ice. After homogenization, the isolated nuclei were washed twice with wash buffer. The fresh single nuclei were fixed, permeabilized, and qualified according to cell line protocol, and then processed to single nuclei RNA-seq according to the following snRandom-seq protocol.

### Technical replication experiment

Two identical samples of FFPE mouse kidney were separately cut from the same paraffin block, and then processed with the snRandom-seq protocol.

### In situ DNA block

The qualified FFPE nuclei were counted and a total of 100,000–1,000,000 nuclei were used for in situ DNA blocking. The following reaction mix was prepared: nuclei in 25.5 μL PBS, 5 μL 10 μM block primer, 2 μL DNA Polymerase, 10 μL 5X DNA polymerization buffer, 5 μL 100 mM dNTP, 2.5 μL RNase Inhibitor. The sequence of block primer was provided in the Supplementary Information file (Supplementary Table 1: Primers). The DNA Polymerase kit was included in the VITAPilote-EFT1300 kit (Cat # R20123124) ordered from M20 Genomics. The reaction mix was incubated at 37 °C for 30 min. After incubation, nuclei were washed with PBST (1X PBS with 0.05% T-ween 20) three times to wash away the residual blocking primers and primer dimers.

### In situ reverse transcription

In situ, reverse transcription was performed using the following reaction mix: 100,000–1,000,000 nuclei in 22.5 μL PBS, 5 μL 10 μM random primer, 5 μL 10 μM oligo(dT) primer, 2.5 μL Reverse Transcriptase, 10 μL

5X reverse transcription buffer, 2.5 μL 100 mM dNTP, 2.5 μL RNase Inhibitor. The sequences of random primer and oligo(dT) primer were provided in the Supplementary Information file (Supplementary Table 1: Primers). The reverse transcription kit was included in the VITAPilote-EFT1300 kit (Cat # R20123124) ordered from M20 Genomics. The reaction mix was incubated with twelve cycles of multiple annealing ramping from 8 °C to 42 °C and 30 min at 42 °C. After reverse transcription, nuclei were washed with PBST three times to wash away the residual random primer and primer dimers.

### dA tailing

For dA tailing, the following reaction mix was prepared: 100,000–1,000,000 nuclei in 39 μL PBS, 5 μL 10X TdT reaction buffer, 0.5 μL TdT enzyme, 0.5 μL 100 mM dATP (NEB, Cat # N0440S), 5 μL CoCl₂. The TdT reaction kit was ordered from NEB (Cat # M0315S), including 10X TdT reaction buffer, TdT enzyme, and CoCl₂. The dA tailing reaction mix was incubated at 37 °C for 30 min and then washed with PBS with 0.05% Tween 20 three times.

### Microfluidic device fabrication

Microfluidic devices were designed using AutoCAD (2021, AutoDESK, USA) according to our previous work[17]. Computer Assisted Designs were printed as photomasks to solidify a raised pattern as a master on a silicon wafer. The device design is provided in Supplementary 1a and Supplementary 2a. The channel depth on devices for hydrogel beads is 30 μm, and for cell encapsulation is 50 μm. Microfluidic devices were fabricated using polydimethylsiloxane (PDMS) according to the protocol described[42]. PDMS base and curing agents (10:1, wt/wt) were mixed by Thinky Mixer and marked into channels using the master as a mold. Then a PDMS slab was acquired, and the inlet and outlet ports were punched. The channel side was treated with oxygen plasma and the PDMS slab was bonded with a glass slide to obtain the microfluidic device. Dealt the channel surfaces with perfluorododecyltrichlorosilane for fluorophilic coating to produce monodisperse and reliable droplets before using this device.

### Barcode beads synthesis

We designed the barcoded beads based on previous work[16,17] and customized them with M20 Genomics company. Hydrogel beads were synthesized by the microfluidic emulsification and polymerization of the acrylamide-primer mix. The acrylamide-primer mix contains 1× acrylamide:bis-acrylamide solution (Invitrogen, Cat #AM9022), 50 μM acrydite-modified oligonucleotides, 10% wt/vol ammonium persulfate (APS, Sangon Biotech Cat #A100486-0025), and 1×Tris-buffered saline-EDTA-Triton (TBSET) buffer. In this study the acrydite-modified oligonucleotide was designed to contain a deoxy Uridine base, instead of a photocleavable moiety. Then the beads were split into a 96-well plate within unique barcode primers for 3-step ligations instead of 2-step extension reactions. The sequences of barcode primers were provided in the Supplementary Information file (Supplementary Table 1: Primers).

### Droplet barcoding

The droplet barcoding was performed according to previous work[16,17]. The morphology of single nuclei after in situ reactions was observed by optical microscope. Single nuclei were counted and diluted with a 30% density gradient solution. Nuclei, 2X DNA extension reaction mix and barcoded beads were encapsulated into droplets using the microfluidic platform as previously described. The 2X DNA extension reaction mix was ordered from M20 Genomics. Then, the emulsions were incubated at 37 °C for 1 h, 50 °C 30 min, 60 °C 30 min, and 75 °C 20 min. After the barcoding reaction, droplets were broken by mixing with PFO buffer. The aqueous phase was taken out and purified by Ampure XP beads (Beckmen, Cat #A63881). PCR was performed to amplify the purified product with Primer1 and Primer2 primers

(Supplementary Table 1). The amplified product was purified by Ampure XP beads and quantified by Qubit (Invitrogen).

### Library preparation

After amplification and purification, VAHTS Universal DNA Library Prep Kit for Illumina V3 (Vazyme, Cat #ND607-01) was used to construct library. The input-DNA was quantified by Qubit2.0 (Life Technologies), and the size was measured with Qsep100™ DNA Fragment Analyzer (BIOptic). Then end-repair and adenylation were performed. The reaction mixture containing fragmented DNA (50 ng), end repair buffer, end repair enzymes, and nuclease-free water was incubated at 30 °C for 30 min and inactivated at 65 °C for 30 min. The finished end-prep reaction mixture was added with working adaptor and ligation enzymes and then was incubated at 20 °C for 15 min. The ligated DNA was purified, and size was selected with AMPure XP beads (Beckmen, Cat #A63881). The library amplification was followed, and purification was performed with AMPure XP beads. The final library was quantified by Qubit2.0 and the library size was measured with Qsep100™ DNA Fragment Analyzer. Library sequencing was performed using the NovaSeq 6000 and S4 Reagent Kit with paired-end reads of 150.

### Data analysis

**Preprocessing of snRandom-seq data.** First, primer sequences and extra bases generated by the dA-tailing step were trimmed in raw sequencing data. Then for each Read1, we extracted UMI (8 nts) and cell-specific barcode (30 nts) and merged sequenced barcodes that can be uniquely assigned to the same accepted barcode with a Hamming distance of 2 nts or less. Read2 was used to generate the gene expression matrix by the STARsolo module in STAR (2.7.10a) with reasonable parameters. The valid nuclei were identified by STARsolo. Bedtools (2.26.0) was used to calculate transcriptome coverage. IGV(2.13.2) was used to generate genome coverage plot. ggplot2 (3.3.5) in R (4.2.1) was used to generate raw plots.

**Clustering and downstream analysis.** The barcode-filtered gene expression matrix was generated with mitochondrial RNAs and ribosomal RNAs removed. snRNA-seq data analysis and visualization were done using RStudio and Seurat 3 toolkit, including preprocessing, integration, visualization, clustering, cell type identification, differential expression testing. Nuclei with less than 200 detected genes and genes detected in less than three nuclei were filtered out. For snRNA-seq datasets integration, counts were first normalized using sctransform function in Seurat[43] and integrated using canonical correlation analysis (CCA)[44]. Integrations were performed across the FFPE/fresh comparison samples (FFPE1, FFPE2, and fresh). For each sample, 2000 anchors were identified, and snRNA-seq datasets were integrated with the IntegrateData function using 20 dimensions[44]. The integrated datasets were constructed the shared nearest neighbor (SNN) graph by running principal component analysis (PCA), FindNeighbors with 30 PCs, FindClusters function with a resolution of 1. Clusters were visualized using t-distributed Stochastic Neighbor Embedding (t-SNE) of the principal components as implemented in Seurat. The cell-type identities for each cluster were determined manually using published list of marker genes. Marker genes were identified test using the FindAllMarkers function in Seurat and kept marker genes matching the filter criteria (only.pos = TRUE, min.pct = 0.25, logfc.threshold = 0.25). Cell-cycle phases were predicted using a function included in Seurat that scores each cell based on expression of canonical marker genes for S and G2/M phases.

**Correlation analysis.** To compare the gene expression level between FFPE and fresh sample, as well as between the two technical replicates, we imported the count data into Seurat (v3 and v4.1.1), normalized, and scaled the data with the default settings. Then we calculated the average normalized expression using Seurat's AverageExpression

function. After that, we plotted the natural logarithm of the average expression with one added pseudo count and calculated the coefficient of variation and *p*-value using ggpubr (0.4.0) in R(4.2.1).

**RNA velocity analysis.** The output files of snRandom-seq data were processed with the scVelo (version 0.2.4) to tag spliced and unspliced transcripts, and the results were analyzed with the velocyto.R 0.6 package in R.

**Cell interaction analysis.** Cell communication analysis was performed using the R package CellChat (version 1.6.1) with default parameters.

### Statistics and reproducibility

Statistical details for each experiment are provided in the figure legends. The microfluidic encapsulation experiment, FFPE single nuclei isolation experiment, beads synthesis experiment, DNA fragments analysis experiment, proteinase K and collagenase comparison experiment, and RNA quality (DV200) comparison experiment were repeated more than five times independently with similar results. The 293T-3T3 mixture experiment and DNA block experiment were repeated three times independently with similar results. No statistical method was used to predetermine the sample size. No data were excluded from the analyses. The experiments were not randomized. The Investigators were not blinded to allocation during experiments and outcome assessment.

### Reporting summary

Further information on research design is available in the Nature Portfolio Reporting Summary linked to this article.

## Data availability

The snRandom-seq snRNA-seq datasets generated in this study have been deposited in the Genome Sequence Archive under accession code "CRA010745" (293T-3T3 mixture and mouse FFPE tissues) and "HRA003712" (MTM-HCC and normal HCC FFPE samples). The public scRNA-seq data of 293T and 3T3 cell mixture by 10X Chromium Single Cell 3' Solution V3 used in this study are available in the Short Read Archive under accession code "SRP073767". The public scRNA-seq data of 293T cells by VASA-drop is available at the Gene Expression Omnibus under accession code "GSE176588". Source data are provided in this paper.

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

## Acknowledgements

The project was supported by the National Natural Science Foundation of China (No. 32200073, Y.W., No. 82200977, Z.X.) and Leading Innovative and Entrepreneur Team Introduction Program of Zhejiang (No. 2021R01012, Y.W.). Thanks for the technical support by the core facilities of Zhejiang University Medical Center and Liangzhu Laboratory. We thank Jingyao Chen and Chengcheng Zhang from the Core Facilities, Zhejiang University School of Medicine for their technical support.

## Author contributions

Y.W. and Z.X. conceived the study and wrote the paper. Y.W., G.G., and J.L. designed the project. Z.X., Y.Z., S.Z., H.D.C., Y.L., H.Y., and L.D. developed the snRandom-seq chemistry. Z.X., T.Z., L.F., H.Y.C., J.C., S.N., F.L., Z.W., D.Z., and Y.C. analyzed the data. Y.W., S.Z., and Y.L. constructed the microfluidic platform. L.Y., W.J., F.C., and H.Z. contributed to the FFPE human experiment. Y.W., G.G., and L.F. supervised this project. All authors have revised and approved the final manuscript.

## Competing interests

Y.W. and Z.X. have submitted a patent application to the Chinese patent office pertaining to the method of this work (application number: CN 114507711 A). Multiple authors are involved in commercialization of the technique and engage with M20 Genomics, Inc. (L.D. and J.L. are co-founders and employees; Y.W. and G.G. are co-founders, equity holders, and consultants; T.Z. and S.N. are employees.) The remaining authors declare no competing interests.
