## [Peer Review File · Nature Communications]

REVIEWER COMMENTS

Reviewer #1 (Remarks to the Author):

Xu et al developed snRandom-seq technology and proved the technology itself works. It is highly needed technology in the field, and could have a broad application in clinically archived samples. However, major flaws are from both technology itself and application of this technology.

1) They used human-mouse mixture cell lines to benchmark their technology, which is not a proper choice. Since the fixation condition in their cell line is very different from FFPE. The purpose of mixing cells from different species could be for accuracy of droplet but not for the FFPE sample itself. Does this part really need?

2), The quality control (benchmark) should be comparison of frozen sample and FFPE sample with same technology such as their figure 4. Before they could prove their single cell profile from FFPE samples are similar as profile got from frozen sample, I did not see the point to directly compare their result with the other technology in Figure 3.

3), In Figure 4b, author claimed the correlation of FFPE and Frozen RNA profile is $R = 0.91$. Could they show a merged genome browser track? How reproducible of this data? It is well known: RNA is very fragile, and the procedure of FFPE block preparation is very harsh. It is very logic to expect the RNA degradation during FFPE preparation. In the field, if we just perform RNA-seq in cell lines not in FFPE samples, what is the potential correlation between duplicates from RNA-seq library? Usually it should be lower than 0.91. Does the good correlation ($R = 0.91$) mean their RNA quality from frozen samples is as poor as FFPE sample?

4), In Figure 4e, it is very impressive they catch so many different cell types. It is very logic to put tSNE plot of single cell RNA-seq from frozen samples with same technology, side by side with FFPE-kidney. The cell components captured in frozen sample are benchmark for FFPE-samples.

5, Figure 5, could they learn something more except of cell types? If just want to learn cell types, why do people need work on FFPE sample? e.g, it would be very valuable to perform on tumor relapse samples, since relapse is common in human tumor and most of those samples are in FFPE format.

6), The nuclei isolation step is very simple and short but also very impressive (from DAPI staining). Proteinase K was added into the lysis buffer, incubating at 37°C for only 5 mins, such nuclei isolation step is quite short and mild comparing with published or BioRxiv papers, Where Collagenase is often used to digest only extracellular matrix but not protein inside cells or nuclei. Proteinase K digestion is very rarely used to isolate nuclei for DNA or RNA studies, such enzyme could enter nuclei and digest proteins inside of nuclei. Is there potential sub-selection of subpopulation cells with such mild digestion condition? Could such enzyme digest RNA binding proteins to further destroy RNA? A detail comparison digestion with Protein K and collagenase would be very useful.

Reviewer #2 (Remarks to the Author):

In this report, the authors developed a new protocol, snRandom-seq, for performing snRNA-seq on FFPE samples. Comparing with current methods, snRandom-seq shows significantly improved performance. Instead of 3' end, full length RNA is covered. In addition, total RNA instead of PolyA RNA species can be captured. The number of transcript and UMI captured per nuclei is improved over current method. No significant bias in terms of cell type sampling is observed. Good consistency with fresh sample is achieved. This method has great potential and provides opportunities to profile FFPE samples. The evaluation of the performance of the new protocol is thorough and the result is convincing. My specific comments are the following:

1. As a method paper, it is important to provide great details of the protocol for readers to follow and reproduce the result. The current information provided in the method section does not have sufficient details for others to follow. For example, the volume of the lysis buffer used in the protocol is not provided. I would suggest the authors to provide detailed protocol in supplement file and also put the protocol into protocol.io.
2. It is not clear if the raw data has been deposited into public databases.
3. The doublet rate is the function of number of cells versus number of droplets. Therefore, the low doublet rate is not necessarily an advantage of the snRandom-seq, rather it reflects relatively underloading. Indeed, the number of nuclei obtained is relatively low. I am wondering how many nuclei were loaded to obtain the 2-3000 nuclei. What is the capture rate?
4. I am wondering what is the percentage of reads mapped to the nuclei. Is significant ambient RNA or nuclei with low transcript detected observed?
5. Compared to the fresh tissue, are there any bias in terms of cell proportion? Is snRandom-seq less biased than the two 10x FFPE methods?

Replies to Reviewers' Comments:

Reviewer #1:

Xu et al developed snRandom-seq technology and proved the technology itself works. It is highly needed technology in the field, and could have a broad application in clinically archived samples. However, major flaws are from both technology itself and application of this technology.

1. They used human-mouse mixture cell lines to benchmark their technology, which is not a Proper choice. Since the fixation condition in their cell line is very different from FFPE. The purpose of mixing cells from different species could be for accuracy of droplet but not for the FFPE sample itself. Does this part really need?

Reply:

We thank for the reviewer's comments. snRandom-seq utilizes random primers to capture total RNAs in single nuclei (**Figure 1**), which differs from the current poly(A)-based and probe-based single cell RNA-seq methods. Therefore, the standard mixed species experiment usually is necessary to evaluate the fidelity of the random primer-based chemistry in snRandom-seq.

Figure 1: snRandom-seq for FFPE tissues overview

The following sentences have been added into the revised manuscript (Page 5):

“snRandom-seq utilizes random primers to capture total RNAs in single nuclei (Fig. 1), which differs from the current poly(A)-based and probe-based single cell RNA-seq

methods. Therefore, we performed a standard mixed species experiment with cultured human (293T) and mouse (3T3) cell lines to assess the fidelity of snRandom-seq.”

2. The quality control (benchmark) should be comparison of frozen sample and FFPE sample with same technology such as their figure 4. Before they could prove their single cell profile from FFPE samples are similar as profile got from frozen sample, I did not see the point to directly compare their result with the other technology in Figure 3.

Reply:

We agree with the reviewer's comments. Accordingly, we have moved the comparison between single cell profiles of fresh sample and FFPE sample from **Figure 4a, b, c, d** to **Figure 3d, e, f, g**, to prove that the single cell profiles from FFPE samples by snRandom-seq are similar with that from fresh samples before comparing snRandom-seq with the other technologies in the revised version.

Figure 3: Comparison of snRandom-seq with other two FFPE snRNA-seq methods

The following sentences have been revised in the manuscript (Page 8):

“To determine whether snRandom-seq can generate enough information from FFPE tissues as fresh samples, we collected both FFPE and fresh samples from the same mouse tissues and compared their RNA profiles using snRandom-seq (**Fig. 3d**). The RNA quality of FFPE and fresh samples were evaluated firstly by the RNA fragments distribution and DV200. As expected, the RNA quality of the FFPE sample was relatively poorer than that of the fresh sample (**Supplementary Fig. 5a**), suggesting that the RNA in the FFPE sample was degraded. The merged genome browser tracks of snRandom-seq results showed that the reads coverage areas of FFPE and fresh samples were overlapped coordinately (**Supplementary Fig. 6a-c**). Consistently, the total RNA profiles of FFPE and fresh samples by snRandom-seq displayed a good correlation (Pearson R: ~ 0.9 , $p < 2.2e-16$; **Fig. 3e**, **Supplementary Fig. 7a, b**). Meanwhile, to prove the repeatability of our method, the same FFPE sample was sequenced independently with snRandom-seq (**Fig. 3d**), and a high correlation (Pearson R ~ 0.92 , $p < 2.2e-16$) of gene expression profiles across these two batches was also seen (**Fig. 3f**). These results showed that snRandom-seq performed well in both fresh and FFPE samples.”

3. In Figure 4b, author claimed the correlation of FFPE and Frozen RNA profile is $R = 0.91$. Could they show a merged genome browser track? How reproducible of this data? It is well known: RNA is very fragile, and the procedure of FFPE block preparation is very harsh. It is very logic to expect the RNA degradation during FFPE preparation. In the field, if we just perform RNA-seq in cell lines not in FFPE samples, what is the potential correlation between duplicates from RNA-seq library? Usually it should be lower than 0.91. Does the good correlation ($R = 0.91$) mean their RNA quality from frozen samples is as poor as FFPE sample?

Reply:

We thank for the reviewer's questions. **1**). We have added the merged genome browser tracks of FFPE and fresh single nuclei RNA profiles (**Supplementary Fig. 6a-c**) in

the revised manuscript. These results showed that the reads coverage areas of FFPE and fresh samples were similar.

Supplementary Figure 6a-c, The whole (a) and split (b, c) merged genome browser tracks visualized by JBrowse.

The following sentences have been added into the revised manuscript (Page 8):

“The merged genome browser tracks of snRandom-seq results showed that the reads coverage areas of FFPE and fresh samples were similar (Supplementary Fig. 6a-c).”

2). To demonstrate the repeatability, we added the correlation of FFPE and fresh testis single nuclei RNA profiles ($R=0.87$) (Supplementary figure 7a) and the correlation of FFPE and fresh heart single nuclei RNA profiles ($R=0.94$) (Supplementary figure 7b) in the revised manuscript. These datasets also showed a high correlation.

Supplementary Figure 7a, b, Correlation of the normalized gene expressions between FFPE/fresh mouse testis samples (a) and mouse heart samples (b).

The following sentences have been added into the revised manuscript (Page 8):

“Consistently, the total RNA profiles of FFPE and fresh samples by snRandom-seq displayed a good correlation (Pearson R: ~0.9, $p < 2.2e-16$; Fig. 3e, Supplementary Fig. 7a, b).”

3). We agree that RNA is very fragile and the procedure of FFPE block preparation is very harsh. To evaluate the RNA quality of fresh and FFPE samples, we provided the fragment size distribution and DV200 of RNAs isolated from FFPE and fresh samples in the revised manuscript (Supplementary fig. 5a). The results showed that the fresh sample displayed clearly distinguishable 18S and 28S rRNA peaks and high DV200 rate (89.8%), which suggests that the fresh sample used in this study has high RNA quality. As expected, the RNA quality of the FFPE sample (DV200: 72.5%) was lower than that of the fresh sample. snRandom-seq uses random primers to capture total RNAs, which makes it less susceptible to RNA degradation. Therefore, snRandom-seq performed well in both fresh and FFPE samples, and the single nuclei RNA profiles of FFPE and fresh samples have a good correlation.

Supplementary Figure 5a, Electropherogram of RNAs isolated from FFPE mouse testis and fresh mouse testis. DV200 value, the percentage of fragments >200 nucleotides.

The following sentences have been added into the revised manuscript (Page 8):

“The RNA quality of FFPE and fresh samples were evaluated firstly by the RNA fragments distribution and DV200. As expected, the RNA quality of the FFPE sample was relatively poorer than that of the fresh sample (**Supplementary Fig. 5a**), suggesting that the RNAs in the FFPE sample was degraded.”

4. In Figure 4e, it is very impressive they catch so many different cell types. It is very logic to put tSNE plot of single cell RNA-seq from frozen samples with same technology, side by side with FFPE-kidney. The cell components captured in frozen sample are benchmark for FFPE-samples.

Reply:

We thank for the reviewer's comments and suggestions. In the revised manuscript, we have added the tSNE plots and cell proportion chart of single cell RNA-seq from fresh and FFPE samples in the **Figure 4d** and supplementary information (**Supplementary Fig 9c**). A robust cell clustering was obtained based on the integrated dataset and t-SNE and most cell types were identified in both fresh and FFPE samples. As expected, there is some difference between FFPE and fresh samples in the proportion of cell types, such as PTC, which might be caused by the sampling error and different nuclei extraction methods for FFPE and fresh samples.

Supplementary figure 9

Figure4

Supplementary Figure 9c, The integrated *t*SNE map contributed by snRandom-seq data from FFPE1, FFPE2 and fresh sample, respectively. Cell types annotated from the integrated snRandom-seq data shown below.

Figure 4d, Proportion of annotated cell types of FFPE1, FFPE2 and fresh samples by snRandom-seq.

The following sentences have been added into the revised manuscript (Page 10):

“By merging the snRandom-seq data of the FFPE samples and fresh sample, as well as the other batch of FFPE samples, we obtained a robust cell clustering by *t*-SNE (*t*-distributed stochastic neighbor embedding) (**Supplementary Fig. 9b**). Most cell types were identified in the three snRandom-seq datasets (**Fig. 4d, Supplementary Fig. 9c**). As expected, there are some differences in the proportion of cell types of the FFPE and fresh samples (such as PTC), which might be caused by the sampling error and different nuclei extraction methods for FFPE and fresh samples.”

5. Figure 5, could they learn something more except of cell types? If just want to learn cell types, why do people need work on FFPE sample? e.g, it would be very valuable to perform on tumor relapse samples, since relapse is common in human tumor and most of those samples are in FFPE format.

Reply:

We thank for the reviewer's comments and suggestions. snRandom-seq can generate gene expression matrixes of thousands of genes and nuclei from FFPE samples. These gene expression matrixes are similar to the current available scRNA-seq datasets and applicable to various scRNA-seq analysis. Accordingly, we have provided more results beside cell types in the revised version. Firstly, for the FFPE sample in Figure 5, we further investigated the cell-cell communication among the clusters and discovered that subpopulations of hepatocytes interacted with other cell clusters through different outgoing and incoming signaling patterns and ligand-receptor pairs (**Fig. 5g-i, Supplementary Fig. 12**). Secondly, we performed snRandom-seq on an additional FFPE specimen (human normal HCC). We found that hepatocyte clusters of normal HCC subtype had markable expression of some lncRNAs

(Supplementary Fig. 13), which demonstrates the advantage of snRandom-seq with full-length transcripts coverage and high sensitivity in non-coding RNA analysis. We agree that performing snRandom-seq on tumor relapse samples would be very valuable. We have designed a following study with more human tumor relapse samples by snRandom-seq and hope to report our results in the future.

Figure 5g. The total number of interactions among different cell populations. **h.** The heatmap showing the outgoing signaling patterns (left) and incoming signaling patterns (right) of each cell cluster. **i.** Bubble diagrams showing the communication probability and statistical significance of receptor-ligand pairs in BMP signaling network.

Supplementary Figure 12a. The inferred BMP signaling pathway network. Heatmap showing the relative importance of each cell cluster based on the computed four network centrality measures of BMP signaling network. **b.** Expression of BMP signaling network-related receptors and ligands.

Supplementary Figure 13. snRandom-seq detected different lncRNAs in subpopulations of human normal HCC FFPE specimens.

a, Histological appearance of normal HCC at low magnification (left, Scale bar, 5 mm.) and high magnification (right, Scale bar, 200 μ m). Red broken line: tumorous area. Blue broken line: sampling area. White broken line: magnified area. **b**, Violin plots and box plots showing the number of genes and UMIs detected in normal HCC FFPE specimens by snRandom-seq. **c**, t-SNE map of nuclei isolated from the FFPE sample based on their gene expressions. Eight cell types, including four hepatocellular subtypes, were annotated and shown. **d**, Top ten markers of each of the eight cell types. Red font: lncRNAs.

The following sentences have been added into the revised manuscript (Page 11, 12): “After further investigating the cell communication among the clusters (Fig. 5g), we found that hepatocyte-1 and hepatocyte-2 displayed different outcoming and

incoming signaling patterns (**Fig. 5h**). Hepatocyte-2 mainly receives signals from plasma cells through the BMP signaling pathway (**Supplementary Fig. 12a**), which is reported to be correlated with tumor progression in HCC^{1, 2}. Ligand-receptor pair analysis found that plasma cells preferentially sent signals to hepatocyte-2 by BMP6-(ACVR1+ACVR2A) and the communication between plasma cells and hepatocyte-2 has specific ligand-receptor pairs, including BMP6-(BMPR1B+BMPR2), BMP6-(BMPR1B+ACVR2B), BMP6-(BMPR1B+ACVR2A), BMP6-(BMPR1A+ACVR2A), and BMP6-(ACVR1+ACVR2A) (**Fig. 5i**). The gene expression also showed that BMPR1B and ACVR2A have specific expressions in hepatocyte-2 (**Supplementary Fig. 12b**). Taken together, snRandom-seq discovered a proliferative and activated subpopulation of hepatocytes from a clinical FFPE specimen, which provides a valuable clue for additional study in future.

SnRandom-seq was also performed on an FFPE specimen of human normal HCC subtype (**Supplementary Fig. 13a**). Based on the snRandom-seq data, sufficient gene count and UMI count were detected, and main cell clusters of the liver were identified (**Supplementary Fig. 13b, c**). Previous studies have indicated that lncRNAs exhibit tissue specific expression^{3, 4}, which are always ignored in routine single cell RNA-seq analysis due to their low expression. We found that hepatocyte clusters of normal HCC subtype had a markable expression of lncRNAs, including LINC02476 and LINC01151 in hepatocyte-2, LINC00540, LINC02307, and LINC02109 in hepatocyte-3, LINC02384 in hepatocyte-4 (**Supplementary Fig. 13d**). It has been reported that LINC02476 promotes the malignant phenotype of HCC by sponging miR-497 and increasing HMGA2 expression⁵, and LINC00540 influences human HCC progression and metastasis via the NKD2-dependent Wnt/ β -Catenin Pathway⁶. These results suggested that hepatocyte-2 (expressed LINC02476) and hepatocyte-3 (expressed LINC00540) of normal HCC subtype might exhibit different pathogenesis. Taken together, snRandom-seq with the advantages of full-length transcripts coverage shows promise in lncRNA analysis in cancer biology.”

6. *The nuclei isolation step is very simple and short but also very impressive (from*

DAPI staining). Proteinase K was added into the lysis buffer, incubating at 37°C for only 5 mins, such nuclei isolation step is quite short and mild comparing with published or BioRxiv papers, Where Collagenase is often used to digest only extracellular matrix but not protein inside cells or nuclei. Proteinase K digestion is very rarely used to isolate nuclei for DNA or RNA studies, such enzyme could enter nuclei and digest proteins inside of nuclei. Is there potential sub-selection of subpopulation cells with such mild digestion condition? Could such enzyme digest RNA binding proteins to further destroy RNA? A detail comparison digestion with Proteinase K and collagenase would be very useful.

Reply:

We thank for the reviewer's comments and suggestions. **1).** We agree that proteinase K digestion is very rarely used to isolate nuclei for DNA or RNA studies. We found that it is hard to isolate intact and clean single nuclei from FFPE samples due to the crosslink of proteins, DNAs, and RNAs in FFPE samples by fixation. Therefore, we chose the broad-spectrum protease, proteinase K, to digest the cross-linked proteins and isolate clean single nuclei from the FFPE samples.

2). Our single nuclei isolation condition includes mechanical lysis (homogenization), chemical lysis (non-ionic surfactant) and enzyme lysis (proteinase K). This digestion condition can isolate clean single nuclei from multiple different tissues by the three lysis processes working together (**Supplementary Fig. 4a**), and nearly all cell types were identified in the single nuclei RNA-seq results by snRandom-seq (**Fig. 4a**)

Supplementary Figure 4

Figure 4

Supplementary Figure 4a, DAPI images of nuclei isolated from FFPE mouse tissues, including heart, liver, testis, and an about 2-year-old FFPE human liver cancer. Scale bar, 50 μ m.

Figure 4a, *t*-SNE analysis of nuclei isolated from FFPE mouse kidney sample by snRandom-seq based on their gene expressions and colored by identified cell types. Fourteen Cell types identified were colored and shown below.

3). We chose a short incubation time of proteinase K to avoid digesting RNA-binding proteins to further destroy RNA. High-quality cDNA libraries were generated by snRandom-seq from multiple FFPE tissues (**Fig. 3c**, **Supplementary fig. 4c, d**), which suggested that the RNAs in the isolated single nuclei were not destroyed by proteinase K.

Figure 3

Supplementary Figure 4

Figure 3c, Electropherogram of FFPE mouse kidney cDNA library for Qsep100™ DNA Fragment Analyzer. Lower marker: 20 bp; upper marker: 1k bp.

Supplementary Figure 4c, d, Electropherogram of cDNA libraries from FFPE mouse

liver (c), FFPE human liver cancer (d). Lower marker: 20 bp; upper marker: 1k bp.

4). We added a comparison of Proteinase K and collagenase (**Supplementary Fig. 3a**). The results showed that digestion with Proteinase K could isolate cleaner single nuclei from FFPE samples.

Supplementary Figure 3a, DAPI images of nuclei isolated from FFPE mouse kidney by Proteinase K (0.5 mg/ml) and Collagenase (1.5 mg/ml). Scale bar, 50 µm.

The following sentences have been added into the manuscript (Page 7): “For FFPE tissues, digestion with Proteinase K could isolate cleaner single nuclei than with collagenase (**Supplementary Fig. 3a**). ”

Reviewer #2

In this report, the authors developed a new protocol, snRandom-seq, for performing snRNA-seq on FFPE samples. Comparing with current methods, snRandom-seq shows significantly improved performance. Instead of 3' end, full length RNA is covered. In addition, total RNA instead of PolyA RNA species can be captured. The number of transcript and UMI captured per nuclei is improved over current method. No significant bias in terms of cell type sampling is observed. Good consistency with fresh sample is achieved. This method has great potential and provides opportunities to profile FFPE samples. The evaluation of the performance of the new protocol is thorough and the result is convincing. My specific comments are the following:

1. As a method paper, it is important to provide great details of the protocol for readers to follow and reproduce the result. The current information provided in the method section does not have sufficient details for others to follow. For example, the volume of the lysis buffer used in the protocol is not provided. I would suggest the authors to provide detailed protocol in supplement file and also put the protocol into protocol.io.

Reply:

We thank for the reviewer's comments and suggestions. We have provided a detailed protocol in the revised supplement file (Supplementary Protocol: snRandom-seq protocol 1.0) and we will put the protocol into protocol.io once this paper is published. The detailed protocol includes the volume of lysis buffer and permeabilization buffer, reaction systems, and reaction programs.

The following sentences have been added into the manuscript (Page 16): “A detailed protocol, including the volume of the lysis buffer and permeabilization buffer, reaction systems, and reaction programs, was provided in the Supplement file (Supplementary Protocol: snRandom-seq protocol 1.0).”

2. It is not clear if the raw data has been deposited into public databases.

Reply:

We thank for the reviewer's suggestions. We are depositing the raw data of this study into the public databases, and the accession codes will be available before publication.

The following sentences have been added into the revised manuscript (Page 20):

“Data Availability

The public scRNA-seq data of 293T and 3T3 cell mixture by 10X Chromium Single Cell 3' Solution V3 is available in the Short Read Archive under accession number SRP073767 and scRNA-seq data of 293T cells by VASA-drop is available at the Gene Expression Omnibus under accession number GSE176588. The snRandom-seq scRNA-seq datasets by this study have been deposited in the Gene Expression Omnibus. Source data are provided with this paper.”

3. The doublet rate is the function of number of cells versus number of droplets. Therefore, the low doublet rate is not necessarily an advantage of the snRandom-seq, rather it reflects relatively underloading. Indeed, the number of nuclei obtained is relatively low. I am wondering how many nuclei were loaded to obtain the 2-3000 nuclei. What is the capture rate?

Reply:

We thank for the reviewer's questions. **1).** We agree that the doublet rate is the function of the number of cells versus the number of droplets. The low doublet rate in snRandom-seq benefits by the pre-indexing of entire transcriptomes using pre-indexed primers during reverse transcription step, which is similar to the published scifi-RNA-seq⁷. We added the species-mixing scatter plot of the human-mouse mixture sample without pre-indexed random primers (**Supplementary figure 1c**). These results showed that the pre-indexed random primers of snRandom-seq significantly decreased the doublet rate (2.9% to 0.3%).

Supplementary figure 1

Figure 2

Supplementary Figure 1c, Species-mixing scatter plot shows the single-nuclei capture efficiency and doublet rate of snRandom-seq without pre-indexed random primers.

Figure 2e, Species-mixing scatter plots showing the single-nuclei capture efficiency and doublet rate of snRandom-seq.

The following sentences have been added into the revised manuscript (Page 5):

“To decrease the doublet rate, we involved a pre-indexing strategy into the reverse transcription step according to the published scifi-RNA-seq⁷. The nuclei were split into different tubes for reverse transcription with pre-indexed random primers, then pooled for the subsequent reaction.”

The following sentences have been added into the revised manuscript (Page 6):

“We counted the ratio of reads mapped to both human and mouse genomes in every single nucleus and found that pre-indexed primers markedly decreased the doublet rate (from 2.9% to 0.3%) (Fig. 2e, Supplementary 1c).”

2). In Figure 3, we sequenced and analyzed 19,258 single nuclei from four FFPE mouse tissues (kidney: 4,716, liver: 3,989, testis: 6,683, and heart: 7,146) using snRandom-seq (Figure 3h). The numbers of nuclei obtained by us are comparable with current other public FFPE single nuclei RNA-seq data. We loaded 6,000~10,000 nuclei to obtain 2~3,000 nuclei. The capture rate of snRandom-seq is 20~50%. The capture rate for human-mouse mixture experiment is 42.2%.

Figure 3h, Gene detection comparison of mouse tissues (heart, kidney, testis and liver) and human liver using snRandom-seq with mouse brain by snFFPE-seq⁸ and breast by snPATHO-seq⁹.

The following sentences have been added into the revised manuscript (Page 6):

“The nuclei capture rate was 42.2% and the percentage of reads mapped to the true nuclei was 76%.”

4. I am wondering what is the percentage of reads mapped to the nuclei. Is significant ambient RNA or nuclei with low transcript detected observed?

Reply:

We thank for the reviewer's questions. The percentage of reads mapped to the nuclei in **Figure 2d** is 76%. Ambient RNA or nuclei with low transcript were observed in the barcode-gene rank plot by snRandom-seq (**Figure 2d**). The significant steep slope in the plot suggested a clear separation of true nuclei from background noise. We used STARsolo to automatically determine the filtering threshold (the dot line) to separate true nuclei and background noise.

Figure 2d, Barcode plot for identification of the barcodes that represent true nuclei (red line). Barcodes of the 293T-3T3 mixed nuclei were ordered from the largest to smallest gene counts.

The following sentences have been revised in the manuscript (Page 6): “After data processing, we identified 2,283 high-quality unique nucleus barcodes by the significant steep slope in the barcode-gene rank plot (Fig. 2d), which suggests a clear separation of true nuclei from background noise. The nuclei capture rate was 42.2% and the percentage of reads mapped to the true nuclei was 76%.”

5. Compared to the fresh tissue, are there any bias in terms of cell proportion? Is snRandom-seq less biased than the two 10x FFPE methods?

Reply:

We thank for the reviewer's questions. **1).** In the revised manuscript, we have added the tSNE plots and cell proportion chart of single cell RNA-seq from fresh and FFPE samples in the **Figure 4c** and supplementary information (**Supplementary Fig 9c**). A robust cell clustering was obtained based on the integrated dataset and t-SNE and most cell types were identified in both fresh and FFPE samples. As expected, there is some difference between FFPE and fresh samples in the proportion of cell types, such as PTC, which might be caused by the sampling error and different nuclei extraction methods for FFPE and fresh samples.

Supplementary figure 9

Figure4

Supplementary Figure 9c, The integrated tSNE map contributed by snRandom-seq data from FFPE1, FFPE2 and fresh sample, respectively. Cell types annotated from the integrated snRandom-seq data shown below.

Figure 4d, Proportion of annotated cell types of FFPE1, FFPE2 and fresh samples by snRandom-seq.

The following sentences have been added into the revised manuscript (Page 10):

“By merging the snRandom-seq data of the FFPE samples and fresh sample, as well as the other batch of FFPE samples, we obtained a robust cell clustering by *t*-SNE (*t*-distributed stochastic neighbor embedding) (Supplementary Fig. 9b). Most cell types were identified in the three snRandom-seq datasets (Fig. 4d, Supplementary Fig. 9c). As expected, there are some differences in the proportion of cell types of the FFPE and fresh samples (such as PTC), which might be caused by the sampling error and different nuclei extraction methods for FFPE and fresh samples.”

2). We believe that snRandom-seq is less biased than the two 10X FFPE methods (snPATHO-Seq and snFFPE-seq). 10X Chromium fixed RNA profile method limits to specific genes and 10X Chromium V3 method limits to poly(A)⁺ RNAs, while snRandom-seq is applicable across all species and genes. In addition, the results of genic coverage indicated that snRandom-seq has more homogeneous distribution across the gene body and higher coverage than the other two 10X FFPE methods (Fig. 2j-l).

Figure 2j, Reads distribution along the gene body by three different snRNA-seq methods (snRandom-seq, snFFPE-seq⁸ and 10X Chromium Fixed RNA Profiling). **k**, Histogram showing the gene body coverage percentages datasets generated by the three methods. **l**, Representative raw reads aligned to human gene *C1S* in snRandom-seq and 10X Chromium Fixed RNA Profiling.

References

1. Wang, Y. et al. LncRNA HAND2-AS1 promotes liver cancer stem cell self-renewal via BMP signaling. *EMBO J.* **38**, e101110 (2019).
2. Ning, J. et al. Imbalance of TGF- β 1/BMP-7 pathways induced by M2-polarized macrophages promotes hepatocellular carcinoma aggressiveness. *Mol. Ther.* **29**, 2067-2087 (2021).
3. Cabili, M.N. et al. Integrative annotation of human large intergenic noncoding RNAs reveals global properties and specific subclasses. *Genes Dev* **25**, 1915-1927 (2011).
4. Liu, S.J. et al. Single-cell analysis of long non-coding RNAs in the developing human neocortex. *Genome Biol.* **17**, 67 (2016).
5. Duan, Y., Zhao, M., Jiang, M., Li, Z. & Ni, C. LINC02476 Promotes the Malignant Phenotype of Hepatocellular Carcinoma by Sponging miR-497 and Increasing HMGA2 Expression. *Onco Targets Ther* **13**, 2701-2710 (2020).
6. Wu, D.-M. et al. Reduced LINC00540 Expression Inhibits Human Hepatocellular Carcinoma Progression and Metastasis via the NKD2-Dependent Wnt/ β -Catenin Pathway. *SSRN Electronic Journal* (2019).
7. Datlinger, P. et al. Ultra-high-throughput single-cell RNA sequencing and perturbation screening with combinatorial fluidic indexing. *Nat. Methods* **18**, 635-642 (2021).
8. Chung, H. et al. SnFFPE-Seq: towards scalable single nucleus RNA-Seq of formalin-fixed paraffin-embedded (FFPE) tissue. *bioRxiv*, 2022.2008.2025.505257 (2022).
9. Vallejo, A.F. et al. snPATHO-seq: unlocking the FFPE archives for single nucleus RNA profiling. *bioRxiv*, 2022.2008.2023.505054 (2022).

REVIEWER COMMENTS

Reviewer #1 (Remarks to the Author):

The authors had revised manuscript. Fundamentally, two concerns are still there in my mind.

1), Technically:

I am still not convinced by their experimental design--- using proteinase K digestion to purify nuclei. Proteinase K could digest any proteins in the nuclei and cells and seriously damage RNA quality. Their argument is: the quality of nuclei from collagenase is not good (revised Supplementary Figure 3). Nuclei purification itself from FFPE tissue itself is not new, and collagenase and many other enzymes are widely used to digest extracellular matrix by protecting intercellular proteins. Here, protein K is used instead, the side effect could be huge.

However, as they had clearly showed: the RNA is highly degraded in Supplementary Figure 5, how the correlation of sequencing libraries between FFPE and frozen is 0.95 (Supplementary Figure 7). As I mentioned, if we compared RNA-seq libraries among frozen technical replication, the range of correlation could be even lower than 0.95.

The revised genome browser track of whole genome scale in Supplementary Figure 6 is useless to compare data quality between FFPE sample and Frozen sample, we could not learn any information of particular genes among different samples.

2), Application:

Single cell or nuclei RNA-seq itself is well established in the field, particularly in frozen sample. The quality of scRNA-seq assay in frozen sample is much better than that from FFPE sample. The strong standing point in the proposed technology is to use FFPE material. The reason to use FFPE is to learn something biological meaning that we could not do in frozen samples. In the revised Figure 5, the added analysis could be performed in frozen samples with better quality.

In my mind, author could easily show advance of their technology by applying in relapse clinical samples or rare disease materials? I think it would be very helpful.

Reviewer #2 (Remarks to the Author):

The authors have provided substantial more detailed information as suggested by the reviewers and did a great job addressing critics. Other than that it seems to me that the detailed protocol supplement file is missing, I feel the manuscript should be accepted.

Reviewer #1 (Remarks to the Author):

The authors had revised manuscript. Fundamentally, two concerns are still there in my mind.

1), Technically:

I am still not convinced by their experimental design--- using proteinase K digestion to purify nuclei. Proteinase K could digest any proteins in the nuclei and cells and seriously damage RNA quality. Their argument is: the quality of nuclei from collagenase is not good (revised Supplementary Figure 3). Nuclei purification itself from FFPE tissue itself is not new, and collagenase and many other enzymes are widely used to digest extracellular matrix by protecting intercellular proteins. Here, protein K is used instead, the side effect could be huge.

Reply: We thank the reviewer's comments. Proteinase K is widely used to digest proteins during RNA or DNA extraction and to inactivate nucleases that could degrade DNA or RNA. Thus, proteinase K itself will not damage RNA quality. Considering the RNA loss due to the digestion of RNA-crosslink proteins by proteinase K, we chose a short incubation time of proteinase K in snRandom-seq to avoid the over digestion inside nuclei. The quality of the FFPE single nuclei isolated by the proteinase K protocol was demonstrated by the clean and intact single nuclei shown in the microscopic images (**Supplementary Fig. 4a**), the acceptable contents of cDNAs showed in the electropherogram of cDNA libraries (**Fig. 3c, Supplementary fig. 4c, d**), the median of >3,000 genes per nucleus detected in various FFPE tissues (**Fig. 3h**), and the complete cell atlas identified in the single nuclei RNA-seq results (**Fig. 4a**). As the reviewer mentioned, nuclei purification from FFPE tissue has been reported in other papers. Martelotto *et al.*¹ used an enzymatic cocktail containing 1 mg/ml of Collagenase/Dispase and 100 units/ml of Hyaluronidase and an incubation of 16 h at 37°C to extract single nuclei from FFPE samples for whole-genome-sequencing. Yadav *et al.*² and Zhang *et al.*³ adopted a similar FFPE nuclei isolation protocol (a collagenase and hyaluronidase cocktail and incubation of 16 h at 37°C) for ATAC-seq. Zhao *et al.*⁴ also adopted a similar FFPE nuclei isolation protocol for antibody-guided chromatin tagmentation with sequencing (FACT-seq). However, these FFPE nuclei isolation protocols were designed for genome sequencing, not for RNA sequencing.

Supplementary Figure 4

Figure 3

Supplementary Figure 4

Figure 3

Figure 4

However, as they had clearly showed: the RNA is highly degraded in Supplementary Figure 5, how the correlation of sequencing libraries between FFPE and frozen is 0.95 (Supplementary Figure 7). As I mentioned, if we compared RNA-seq libraries among frozen technical replication, the range of correlation could be even lower than 0.95.

Reply: Thanks for the reviewer's careful reading. We performed snRandom-seq on three matched pairs of FFPE and fresh/frozen samples. The correlation coefficients of the total RNA profiles of these three pairs of FFPE and fresh/frozen samples were 0.91, 0.88, and 0.95, respectively (Fig. 3e, Supplementary Fig. 7a, b). We found that several previous studies also reported a high correlation between the gene expression profiles of paired FFPE and fresh/frozen samples using different RNA-Seq protocols. For example, Li *et al.*⁵ showed that the fresh frozen and FFPE libraries (FF.K.TotalRNA and FFPE.K.TotalRNA) from human breast cancer tissues using KAPA Stranded RNA-Seq Kit had highly correlated expression levels (correlation coefficient ~0.973). Subsequent exon capture using probes even resulted in a correlation of 0.980 between fresh frozen and FFPE libraries (FF.CR and FFPE.CR). The correlation coefficients among fresh frozen libraries using different protocols (FF.mRNA, FF.I.TotalRNA, and FF.K.TotalRNA) were also higher than 0.95. Bossel *et al.*⁶ reported a high correlation (correlation coefficient ~0.9) of the mRNA-seq whole transcriptome profiling between matched FFPE and fresh frozen tumor samples, with a moderate archival time of about 4-5 years. Cannizzo *et al.*⁷ observed high agreement ($R^2 = 0.954$) in fold changes of differentially expressed genes between paired frozen and FFPE samples stored for over 20 years using targeted resequencing (TempO-Seq). Esteve-Codina *et al.*⁸ also indicated that RNAs

from FFPE specimens were highly degraded by several quality-control measurements but maintained transcriptomic similarities ($R^2 \sim 0.9$) to RNA from matched fresh frozen samples. In addition, MATQ-seq⁹, a single cell total RNA-seq method using random primer, also reported high reproducibility (correlation coefficient: 0.946 ± 0.003) among 6 ERCC spike-in samples.

The revised genome browser track of whole genome scale in Supplementary Figure 6 is useless to compare data quality between FFPE sample and Frozen sample, we could not learn any information of particular genes among different samples.

Reply: We want to thank the reviewer's comment. In addition to the genome browser track of whole genome scale (**Supplementary Fig. 6a-c**), we further enlarged the scale of the genome browser and provided the reads distribution of four randomly selected genes (*Gm37600*, *Pkhd1*, *GM42476*, and *Prex2*) in the merged genome browser track (**Supplementary Fig. 6d-g**). The distribution of mapped reads in the four genic regions indicated a similar pattern, i.e., a highly similar genomic distribution between the FFPE and frozen samples.

Supplementary Figure 6. Comparison of the genome coverage of FFPE and fresh single nuclei RNA profiles by snRandom-seq.

a-c, The whole (a) and split (b, c) merged genome browser tracks visualized by JBrowse. d-g, The

reads distribution of four randomly selected genes (*Gm37600*, *Pkhd1*, *Gm42476*, and *Prex2*) in the merged genome browser track visualized by JBrowse.

2), Application:

Single cell or nuclei RNA-seq itself is well established in the field, particularly in frozen sample. The quality of scRNA-seq assay in frozen sample is much better than that from FFPE sample. The strong standing point in the proposed technology is to use FFPE material. The reason to use FFPE is to learn something biological meaning that we could not do in frozen samples. In the revised Figure 5, the added analysis could be performed in frozen samples with better quality.

In my mind, author could easily show advance of their technology by applying in relapse clinical samples or rare disease materials? I think it would be very helpful.

Reply: Thanks for the reviewer's valuable suggestion. Following the suggestion, we added an application of snRandom-seq on a relapsed case, i.e., a matched pair of initial and relapsed FFPE clinical specimens from the same colorectal cancer liver metastasis (CRLM) patient. The initial FFPE specimen was sampled at the initial diagnosis and archived for over two years. The relapsed FFPE specimen was sampled from the same patient with relapsing and archived for about one year. snRandom-seq detected medians of ~1,000 gene counts and ~2,000 UMI counts in both initial and relapsed FFPE specimens (**Supplementary Fig. 14a**). The cells from the initial and relapsed FFPE specimens were comprehensively integrated, and the major cell types (hepatocytes, cancer cells, T cells, fibroblasts, myeloid cells, endothelial cells, stellate cells, macrophages, cholangiocytes, B/plasma cells) were identified in both samples (**Supplementary Fig. 14b, c**). We observed that the proportion of T cells was higher in the relapsed FFPE sample (**Supplementary Fig. 14d**), suggesting a more active antitumor immune response in the relapsed sample. Consistently, the proportions of the dominating cancer clusters (cancer cells-1, -2, and -3) were decreased in the relapsed sample (**Supplementary Fig. 14d**). However, the proportion of cancer cells-4 was increased in the relapsed sample (**Supplementary Fig. 14d**). We further found that the genes encoding lipids composition regulator (*SCD*) and proteins binding lipids (*APOA2*, *APOC3*, and *APOA1*) displayed high expression levels in the cancer cell-4 cluster in the relapsed sample (**Supplementary Fig. 14e**), suggesting an enhanced lipid metabolism in the cancer cells subcluster of the relapsed CRLM.

Supplementary Figure 14. snRandom-seq revealed cell heterogeneity of initial and relapsed FFPE specimens.

a, Violin plots and box plots showing the number of genes and UMIs detected in initial and relapsed FFPE specimens from a same colorectal cancer liver metastasis patient by snRandom-seq. **b-c**, *t*-SNE map of integrated nuclei from initial and relapsed FFPE specimens colored by groups (initial, relapsed) (**b**) and cell types (hepatocytes, cancer cells, T cells, fibroblasts, myeloid cells, endothelial cells, stellate cells, macrophages, cholangiocytes, B/plasma cells) (**c**). **d**, Percent of cells in the different cell types. **e**, Expression levels of lipid metabolism related genes (*SCD*, *APOA2*, *APOC3*, and *APOA1*) in cancer cells.

Reviewer #2 (Remarks to the Author):

The authors have provided substantial more detailed information as suggested by the reviewers and did a great job addressing critics. Other than that it seems to me that the detailed protocol supplement file is missing, I feel the manuscript should be accepted.

Reply: We are grateful for the reviewer's effort to review our paper with positive feedback. We are sorry for the miss and provided the detailed protocol supplement file (**Supplementary Protocol: snRandom-seq protocol 1.0**).

References

1. Martelotto, L.G. et al. Whole-genome single-cell copy number profiling from formalin-fixed paraffin-embedded samples. *Nat Med* **23**, 376-385 (2017).
2. Yadav, R.P., Polavarapu, V.K., Xing, P. & Chen, X. FFPE-ATAC: A Highly Sensitive Method for Profiling Chromatin Accessibility in Formalin-Fixed Paraffin-Embedded Samples. *Current Protocols* **2**, e535 (2022).
3. Zhang, H. et al. Profiling chromatin accessibility in formalin-fixed paraffin-embedded samples. *Genome Res.* **32**, 150-161 (2022).
4. Zhao, L. et al. FACT-seq: profiling histone modifications in formalin-fixed paraffin-embedded samples with low cell numbers. *Nucleic Acids Res.* **49**, e125 (2021).
5. Li, J., Fu, C., Speed, T.P., Wang, W. & Symmans, W.F. Accurate RNA Sequencing From Formalin-Fixed Cancer Tissue to Represent High-Quality Transcriptome From Frozen Tissue. *JCO Precision Oncology*, 1-9 (2018).
6. Bossel Ben-Moshe, N. et al. mRNA-seq whole transcriptome profiling of fresh frozen versus archived fixed tissues. *BMC Genomics* **19**, 419 (2018).
7. Cannizzo, M.D., Wood, C.E., Hester, S.D. & Wehmas, L.C. Case study: Targeted RNA-sequencing of aged formalin-fixed paraffin-embedded samples for understanding chemical mode of action. *Toxicology Reports* **9**, 883-894 (2022).
8. Li, P., Conley, A., Zhang, H. & Kim, H.L. Whole-Transcriptome profiling of formalin-fixed, paraffin-embedded renal cell carcinoma by RNA-seq. *BMC Genomics* **15**, 1087 (2014).
9. Sheng, K., Cao, W., Niu, Y., Deng, Q. & Zong, C. Effective detection of variation in single-cell transcriptomes using MATQ-seq. *Nat. Methods* **14**, 267-270 (2017).

REVIEWERS' COMMENTS

Reviewer #1 (Remarks to the Author):

Authors addressed some of my concerns, but the technical concerns in my mind still stand, using Protein K to isolate nuclei. I suggest: authors could use the same protocol (protein K digestion protocol) to isolate nuclei from frozen sample to perform snRandom-seq, by cross-comparison: snRandom-seq Frozen sample with PK digestion v.s snRandom-seq Frozen sample without PK digestion; snRandom-seq FFEP sample v.s snRandom-seq Frozen sample with PK digestion. This cross-comparison will give clear answer: how/whether PK digestion affect RNA quality.

Reviewer #2 (Remarks to the Author):

The authors have addressed the concerns raised by the review sufficiently and the new method has great potential for profiling FFPE samples.

REVIEWERS' COMMENTS

Reviewer #1 (Remarks to the Author):

Authors addressed some of my concerns, but the technical concerns in my mind still stand, using Protein K to isolate nuclei. I suggest: authors could use the same protocol (protein K digestion protocol) to isolate nuclei from frozen sample to perform snRandom-seq, by cross-comparison: snRandom-seq Frzoen sample with PK digestion v.s snRandome-seq Frozen sample without PK digestion; snRandom-seq FFEP sample v.s snRandom-seq Frzoen sample with PK digestion. This cross-comparison will give clear answer: how/whether PK digestion affect RNA quality.

Reply: We thank the reviewer's comments. In this study, snRandom-seq is developed for high-throughput single nuclei total RNA-seq of FFPE samples. We have demonstrated the feasibility of this method for nuclei isolation from FFPE samples.

Reviewer #2 (Remarks to the Author):

The authors have addressed the concerns raised by the review sufficiently and the new method has great potential for profiling FFPE samples.

Reply: Thanks for the reviewer's comments.